# The Old and the New: Cardiovascular and Respiratory Alterations Induced by Acute JWH-018 Administration Compared to Δ^9^-THC—A Preclinical Study in Mice

**DOI:** 10.3390/ijms24021631

**Published:** 2023-01-13

**Authors:** Beatrice Marchetti, Sabrine Bilel, Micaela Tirri, Raffaella Arfè, Giorgia Corli, Elisa Roda, Carlo Alessandro Locatelli, Elena Cavarretta, Fabio De Giorgio, Matteo Marti

**Affiliations:** 1Section of Legal Medicine and LTTA Center, Department of Translational Medicine, University of Ferrara, 44121 Ferrara, Italy; 2Laboratory of Clinical & Experimental Toxicology, Pavia Poison Centre, National Toxicology Information Centre, Toxicology Unit, Istituti Clinici Scientifici Maugeri IRCCS, 27100 Pavia, Italy; 3Department of Medical-Surgical Sciences and Biotechnologies, Sapienza University of Rome, 00185 Roma, Italy; 4Mediterranea Cardiocentro, 80122 Napoli, Italy; 5Section of Legal Medicine, Department of Health Care Surveillance and Bioetics, Università Cattolica del Sacro Cuore, 00168 Rome, Italy; 6Fondazione Policlinico Universitario A. Gemelli IRCCS, 00168 Rome, Italy; 7Collaborative Center for the Italian National Early Warning System, Department of Anti-Drug Policies, Presidency of the Council of Ministers, 00186 Rome, Italy

**Keywords:** synthetic cannabinoid, cardiovascular, plethysmography, blood pressure

## Abstract

Several new psychoactive substances (NPS) are responsible for intoxication involving the cardiovascular and respiratory systems. Among NPS, synthetic cannabinoids (SCs) provoked side effects in humans characterized by tachycardia, arrhythmias, hypertension, breathing difficulty, apnoea, myocardial infarction, and cardiac arrest. Therefore, the present study investigated the cardio-respiratory (MouseOx Plus; EMKA electrocardiogram (ECG) and plethysmography TUNNEL systems) and vascular (BP-2000 systems) effects induced by 1-naphthalenyl (1-pentyl-1H-indol-3-yl)-methanone (JWH-018; 0.3–3–6 mg/kg) and Δ^9^-tetrahydrocannabinol (Δ^9^-THC; 0.3–3–6 mg/kg), administered in awake CD-1 male mice. The results showed that higher doses of JWH-018 (3–6 mg/kg) induced deep and long-lasting bradycardia, alternated with bradyarrhythmia, spaced out by sudden episodes of tachyarrhythmias (6 mg/kg), and characterized by ECG electrical parameters changes, sustained bradypnea, and systolic and transient diastolic hypertension. Otherwise, Δ^9^-THC provoked delayed bradycardia (minor intensity tachyarrhythmias episodes) and bradypnea, also causing a transient and mild hypertensive effect at the tested dose range. These effects were prevented by both treatment with selective CB_1_ (AM 251, 6 mg/kg) and CB_2_ (AM 630, 6 mg/kg) receptor antagonists and with the mixture of the antagonists AM 251 and AM 630, even if in a different manner. Cardio-respiratory and vascular symptoms could be induced by peripheral and central CB_1_ and CB_2_ receptors stimulation, which could lead to both sympathetic and parasympathetic systems activation. These findings may represent a starting point for necessary future studies aimed at exploring the proper antidotal therapy to be used in SCs-intoxicated patient management.

## 1. Introduction

In the last few decades, the international scene of drugs abuse has faced a challenge due to novel psychoactive substances (NPS). NPS are unregulated, mind-altering chemicals that have become newly available on the market and are intended to produce the same effects as illegal drugs, usually synthesized by clandestine laboratories, working to modify the chemical structures of traditional drugs of abuse or even molecules abandoned by drug development to circumvent laws and control systems [1]. NPS have a powerful pharmaco-toxicological activity causing severe adverse effects, particularly dangerous for consumers’ health [2], and are not recognized by routine tests, resulting in a challenge for clinical and forensic toxicology and for the authorities [3]. In particular, synthetic cannabinoids (SCs) are a growing number of human-made, mind-altering substances that are functionally similar to Δ^9^-THC (Figure 1), the psychoactive component of cannabis, and cause several central and peripheral alterations [4,5].

SCs appeared on the drug market around the mid-2000s as products called “Spice”, and they were sold in smart shops or on internet as herbal mixtures or liquids for e-cigarettes [6,7]. Often, several SCs were contained in a single preparation, leading to a greater risk of overdose and intoxications [4,5,8,9,10]. SCs can promote several behavioral and physiological actions, such as drowsiness, dizziness, hyperemesis, motor impairment, psychomotor agitation, and anxiety, but also psychosis, schizophrenia, and tonic–clonic seizures [2,11,12,13]. The consequences of SCs abuse on human health still remains poorly understood, and even less is known about the medium- and long-term effects of these drugs, including their potential for abuse, dependence, and withdrawal, up to the potential neurotoxicity impact which could induce permanent brain deficits. In fact, neurotoxicity related psychiatric problems and brain abnormalities are relevant, as clearly demonstrated by the epidemiological investigations and post-mortem neuroimaging studies showing evident damage to both neuronal soma and synaptic terminals [14]. Moreover, SCs also affect the cardiovascular and respiratory systems. In particular, SCs induce important changes of blood pressure (hypertension and hypotension), heart rate alterations (tachycardia or bradycardia), and other arrhythmias (e.g., atrial fibrillation, premature beats, prolonged QRS and QT time, repolarization abnormalities, ventricular dysrhythmias, atrio-ventricular blocks) [12,15,16,17,18]. Consequently, ischemic stroke, myocardial infarction, and fatalities due to cardiogenic shock or cardiac arrest have been described [19,20,21].

Beyond cardiovascular alterations, even adverse respiratory effects (e.g., coughing, bradypnea, dyspnea, bronchospasm, hypoxemia, and respiratory insufficiency) were reported in several intoxication cases, in which JWH-018 and analogs have been identified [16,22,23,24]. Furthermore, JWH-018 and its naphthylindol analogues, e.g., JWH-073, JWH-081, JWH-122, and JWH-210, were discovered in patients with tachycardia, bradycardia, chest pain, hypertension, and electrocardiogram abnormalities (e.g., premature beats, ST elevation, QT prolongation) [15,16].

Concerning preclinical studies, SCs have been demonstrated to provoke cardiovascular and respiratory toxicity in rats. In particular, bradycardia, vasoconstriction, and increased blood pressure have been shown with different onset and extent after administration of diverse SCs, e.g., JWH-018 [25], other naphthylindole derivates [26], and other SCs or AKB-48 [27,28,29,30,31]. In regard to respiratory effects, CP55, 940, WIN55212-2, and AKB-48 have been demonstrated to decrease the respiratory rate and induce hypoxia [27,28].

All these effects were mainly prevented by CB_1_-selective antagonists AM 251 [28,30] or SR141716A [27], demonstrating that these effects are primarily due to CB_1_ receptor activation. Nevertheless, there is evidence that the CB_2_ receptors may also be involved in cardiovascular [32] and respiratory [33] disorders. Beyond cannabinoids substrates, it has to be taken into consideration that SCs can also modulate physiological responses through other receptor targets, such as ion channels, G protein-coupled receptor 55 (GPR55), peroxisome proliferator-activated receptors (PPAR), transient receptor potential channels type vanilloid (TRPV1), and ankyrin (TRPA1) [34,35].

Despite the central modulation of cardiovascular and respiratory functions, SCs can also interact with peripheral CB receptors. Indeed, in humans CB, receptors in the cardiovascular system are localized in different areas. Both the CB_1_ and CB_2_ receptors are expressed in the coronary arteries, myocardium, endothelial, and smooth muscle cells. In particular, CB_1_ receptors are also expressed in pre-synaptic sympathetic nerve terminals innervating the cardiovascular systems [36]. Additionally, the respiratory system is controlled by the peripheral activation of CB_1_ and CB_2_ receptors [37] expressed in airway epithelial cells [38], bronchi, lung tissue, respiratory endothelium [37], and axon terminals of airways nerves [39].

The JWH-018 was the first SC detected in different products and soon recognized to act as a potent and effective CB_1_ receptor agonist able to activate numerous receptors signaling pathways [40]. It was found to bind murine, rat, and human cannabinoid CB_1_/CB_2_ receptors in the low nanomolar range [41], showing approximately a four-fold increased activity at CB_1_ and about a ten-fold increased affinity at CB_2_ receptors, compared with Δ^9^-THC [41,42]. As previously reported in the literature about other SCs, JWH-018 caused acute toxic effects in users, such as headache, nausea, diaphoresis, psychomotor agitation, seizures, panic attacks, paranoia, psychosis, hallucinations, and cardiovascular and respiratory symptoms characterized by palpitations, tachycardia, arrhythmias, hypertension, and respiratory distress [16,42,43,44].

JWH-018 and its naphthoylindole analogs are the most common compounds identified in clinical, toxicological, and forensic analyses after cardiovascular and respiratory acute SCs intoxications [15,16]; thus, an urgent need for further in vivo studies exploring these effects clearly emerged.

Therefore, the current study investigates cardiac (heart rate) alterations and electrocardiogram (ECG) parameters (such as PR interval, QRS complex duration, QT, and QTc interval), together with vascular (pulse distension, systolic, and diastolic pressure) alterations and respiratory (oxygen saturation, breath rate, and plethysmography parameters, such as breathing frequency, breath length, tidal volume, ratio of expired volume/tidal volume, and relaxation time) induced by acute administration of JWH-018 (0.3–3–6 mg/kg) in awake CD-1 male mice. To better evaluate the potency of JWH-018 effects, Δ^9^-THC was used (0.3–3–6 mg/kg) for comparison. Moreover, to better understand how endocannabinoid system is involved in cardiovascular and respiratory changes, CB_1_ (AM 251, 6 mg/kg) and CB_2_ (AM 630, 6 mg/kg) receptor antagonists were administered singularly or co-administered (AM 251 and AM 630), before or after JWH-018 and Δ^9^-THC treatments.

## 2. Results

### 2.1. Heart Rate

The basal heart rate (HR = 680.44 ± 4.5 bpm) did not change in the vehicle-treated mice over the 6-h observation period (Figure 2A,B). The systemic administration of JWH-018 and ∆^9^-THC deeply affected the cardiac activity in mice (Figure 2).

In particular, JWH-018 at a lowest dose (0.3 mg/kg) immediately and transiently reduced the heart rate of mice (up to ~35% of decrement at 30 min from injection), while at higher doses (3 and 6 mg/kg), results showed a long lasting deep bradycardia (decrease of ~40%, after 5 min and 15 min, respectively), that persisted for all 5 h of the experiment, with a peak at 275 min with higher HR (Figure 2A: significant effect of treatment (F_3, 1440_ = 501.7, *p* < 0.0001), time (F_71, 1440_ = 23.64, *p* < 0.0001), and time x treatment interaction (F_213, 1440_ = 3.971, *p* < 0.0001). The JWH-018-induced bradycardia (up to 276.56 ± 5.6 bpm) at highest dose (6 mg/kg) is showed in ECG traces (Figure 3B), which indicated a wider peak-to-peak distance or RR interval (the time elapsed between two successive R-waves of the QRS signal on the electrocardiogram) than the basal ECG (Figure 3A).

In the sole first hours, all JWH-018 doses brought the insurgence of tachyarrhythmia events (Figure 2E; significant effect of treatment (F_3, 196_ = 42.63, *p* < 0.0001), bin (F_6, 196_ = 19.16, *p* < 0.0001) and bin x treatment interaction (F_18, 196_ = 12.67, *p* < 0.0001). In particular, both 0.3 and 3 mg/kg doses induced the onset of about 2000 tachyarrhythmia events during the first hour, while the 6 mg/kg dose induced the insurgence of about 10,000 events, and the effect continued to be relevant for all observation hours (Figure 2G,I,K,M, effect of treatment (F_3, 196_ = 20.21, *p* < 0.0001), bin (F_6, 196_ = 3.104, *p* = 0.0063), and bin x treatment interaction (F_18, 196_ = 3.357, *p* < 0.0001), effect of treatment (F_3, 196_ = 28.36, *p* < 0.0001), bin (F_6, 196_ = 5.106, *p* < 0.0001), and bin x treatment interaction (F_18, 196_ = 2.364, *p* = 0.0021), effect of treatment (F_3, 196_ = 93.03, *p* < 0.0001), bin (F_6, 196_ = 8.106, *p* < 0.0001), and bin x treatment interaction (F_18, 196_ = 8.184, *p* < 0.0001), effect of treatment (F_3, 196_ = 93.03, *p* < 0.0001), bin (F_6, 196_ = 8.106, *p* < 0.0001), and bin x treatment interaction (F_18, 196_ = 8.184, *p* < 0.0001), respectively). Following 6 mg/kg JWH-018 administration, tachycardia and tachyarrhythmia (up to 860.10 ± 5.9 bpm) are highlighted in the ECG traces (Figure 3C,D), which also showed tracks of bradyarrhythmia (Figure 3E). The ∆^9^-THC administration at the highest dose (6 mg/kg) reduced the murine heart rate (~10% during the first hour and ~30% at 2 h from injection; Figure 2B; significant effect of treatment (F_3, 2016_ = 470.8, *p* < 0.0001), time (F_71, 2016_ = 6.592, *p* < 0.0001), and time x treatment interaction (F_213, 2016_ = 4.067, *p* < 0.0001), and these effects were also visible in ECG traces (Figure 3G) that showed a larger RR interval than basal (Figure 3F). On the contrary, the lowest dose of ∆^9^-THC slightly increased the heart rate after two hours from injection. The number of events detected was visibly lower, compared to JWH-018 at all doses. The ∆^9^-THC induced tachyarrhythmias events only at the highest dose (6 mg/kg) administration on the fourth hour of experiment (Figure 2J, effect of treatment (F_3, 196_ = 10.14, *p* < 0.0001), bin (F_6, 196_ = 1.287, *p* = 0.2649), and bin x treatment interaction (F_18, 196_ = 1.288, *p* = 0.1989) and, on a smaller scale, to the fifth and sixth hours (Figure 2L,N effect of treatment (F_3, 196_ = 9.995, *p* < 0.0001), bin (F_6, 196_ = 1.661, *p* = 0.1324), and bin x treatment interaction (F_18, 196_ = 1.773, *p* = 0.0308) and effect of treatment (F_3, 196_ = 21.04, *p* < 0.0001), bin (F_6, 196_ = 10.51, *p* < 0.0001), and bin x treatment interaction (F_18, 196_ = 9.971, *p* < 0.0001), respectively). The ECG trace showed bradyarrhythmia after ∆^9^-THC (6 mg/kg) injection (Figure 3H). Administration of AM 251 (6 mg/kg) did not change the heart rate administered by itself, and it completely and immediately blocked the effect of JWH-018 (6 mg/kg, Figure 4A; significant effect of treatment (F_3, 1728_ = 516.6, *p* < 0.0001), time (F_71, 1728_ = 15.65, *p* < 0.0001), and time x treatment interaction (F_213, 1728_ = 4.847, *p* < 0.0001) and ∆9-THC (6 mg/kg, Figure 4B; significant effect of treatment (F_3, 2016_ = 227.7 *p* < 0.0001), time (F_71, 2016_ = 4.597, *p* < 0.0001), and time x treatment interaction (F_213, 2016_ = 2.803, *p* < 0.0001), respectively) on heart rate.

The administration of CB_2_ selective antagonist, AM 630 (6 mg/kg), and co-administration of both CB_1_ and CB_2_ selective antagonists (AM 630, 6 mg/kg, and AM251, 6 mg/kg, respectively) did not change the heart rate in mice (Figure 5).

The AM 630 administration reverted the effect of JWH-018 after one hour of injection (Figure 5A; significant effect of treatment (F_3, 2016_ = 404.2, *p* < 0.0001), time (F_71, 2016_ = 7.626, *p* < 0.0001), and time x treatment interaction (F_213, 2016_ = 3.930, *p* < 0.0001). After the ∆^9^-THC treatment, AM 630, totally reverted the effect only during the last two hours of experiment (Figure 5B; significant effect of treatment (F_3, 2016_ = 355.4, *p* < 0.0001), time (F_71, 2016_ = 8.743, *p* < 0.0001), and time x treatment interaction (F_213, 2016_ = 3.231, *p* < 0.0001). Co-administration of AM 630 and AM 251 reverted the effects on heart rate of both JWH-018 (Figure 5C; significant effect of treatment (F_3, 2016_ = 437.4, *p* < 0.0001), time (F_71, 2016_ = 5.425, *p* < 0.0001), and time x treatment interaction (F_213, 2016_ = 2.976, *p* < 0.0001) and ∆^9^-THC (Figure 5D; significant effect of treatment (F_3, 2016_ = 243.2, *p* < 0.0001), time (F_71, 2016_ = 2.183, *p* < 0.0001), and time x treatment interaction (F_213, 2016_ = 2.885, *p* < 0.0001) within the first hour from injection.

### 2.2. Electrocardiogram Parameters and Recording

Electrocardiogram parameters have been evaluated after JWH-018 and ∆^9^-THC administration at the highest dose (6 mg/kg, Table 1), relating it to the HR dose–response curve (Figure 2A,B).

Vehicle injection did not affect the basal cardiac electrical parameters in mice in the first and second hours, while during the following hours, a decrease of ~10% on the HR was observed both for the vehicle of JWH-018 (Table 1; significant effect of treatment, t = 42.72, df = 28, *p* < 0.001) and vehicle of ∆^9^-THC (Table 1; significant effect of treatment t = 73.90, df = 28, *p* < 0.001). Consequentially, the RR interval increased of ∼10% (Table 1; significant effect of treatment, t = 42.91, df = 28, *p* < 0.001, and significant effect of treatment t = 22.96, df = 28, *p* < 0.001, respectively).

Related to the JWH-018 HR variation after 6 mg/kg treatment, the decrease is noticeable in the dose–response curve (Figure 2A). ECG parameters showed a significant HR decrease of ∼55% (significant effect of treatment, t = 97.24, df = 28, *p* < 0.001) and a consequent RR increase of ∼95% (significant effect of treatment, t = 52.77, df = 28, *p* < 0.001), together with other electrical parameters changes (Table 1). In particular, QRS complex duration was increased of ∼5% (significant effect of treatment t = 6.245, df = 28, *p* < 0.001), interval QT increased of ∼10% (significant effect of treatment t = 19.67, df = 28, *p* < 0.001), and QTcF that automatically related RR and QT parameters decreased by ∼12% (significant effect of treatment t = 11.93, df = 28, *p* < 0.001) than basal. ECG electrical parameters related to tachyarrhythmias and tachycardia periods, indicated with the letter b, that show the peak of HR increase in JWH-018 treated mice (Figure 2A) were evaluated (Table 1). Comparing them to the basal values, HR increase of ∼16% (significant effect of treatment t = 67.71, df = 28, *p* < 0.001) and RR interval decreased of ∼25% (significant effect of treatment t = 4.160, df = 28, *p* = 0.0003), and the related electrical parameters showed a PR interval increased (∼35%, significant effect of treatment t = 17.56, df = 28, *p* < 0.001), the QT interval increased (∼2%, significant effect of treatment t = 5.015, df = 28, *p* < 0.001), and QTcF increased (∼7%, significant effect of treatment t = 16.14, df = 28, *p* < 0.001). The electrical parameters change after JWH-018 6 mg/kg injection was even more evident when comparing a single ECG wave of vehicle-treated mice with JWH-018-treated animals (Figure 6A).

After ∆^9^-THC treatment (Figure 2B), the HR initially decreased, and the electrical parameters modifications have been shown (Table 1). Together with a HR decrease of ∼13%, compared to basal (significant effect of treatment t = 31.88, df = 28, *p* < 0.001), the ECG values indicated a RR increase of ∼10% (significant effect of treatment t = 17.08, df = 28, *p* < 0.001) and a slight increase of QRS complex duration (∼2% significant effect of treatment t = 2.558, df = 28, *p* = 0.0162). During the last 3 h of experiment, ∆^9^-THC induced a further decrease of HR (Figure 2B), and related to this, the ECG electrical parameters have been evaluated (Table 1). ∆^9^-THC induced a decrease of HR values of ∼22%, compared to basal (significant effect of treatment t = 60.76, df = 28, *p* < 0.001), a RR increase of ∼27% (significant effect of treatment t = 70.43, df = 28, *p* < 0.001), and a slight increase of PR interval (∼2% significant effect of treatment t = 2.178, df = 28, *p* = 0.0380), QRS complex duration (∼3% significant effect of treatment t = 3.168, df = 28, *p* = 0.0037), as well as a slight decrease of QTcF (∼6% significant effect of treatment t = 3.546, df = 28, *p* = 0.014). ECG electrical parameters alteration after ∆^9^-THC 6 mg/kg injection was also manifested when comparing the vehicle ECG wave and ∆^9^-THC ECG wave (Figure 6B).

### 2.3. Pulse Distension

Basal pulse distention (222 ± 17 µm, Figure 7) did not change in the vehicle-treated mice over the 6-h observation period. Administration of JWH-018 (Figure 7A) altered pulse distension in mice (significant effect of treatment (F_3, 1440_ = 97.39, *p* < 0.0001), time (F_71, 1440_ = 3.754, *p* < 0.001), and time x treatment interaction (F_213, 1440_ = 1.357, *p* = 0.0010).

The lowest dose of 0.3 mg/kg of JWH-018 reduced pulse distension during the 2 h after injection (a reduction of ∼42% at 150 min after injection), and 6 mg/kg administration decreased also pulse distension, with a reduction of ∼60% at 215 min after injection. The reduction of basal pulse distention JWH-018 induced was prevented by the treatment with AM 251 (6 mg/kg, i.p; Figure 7C; significant effect of treatment (F_3, 1728_ = 80.58, *p* < 0.0001), time (F_71, 1728_ = 2.746, *p* < 0.0001), and time x treatment interaction (F_213, 1728_ = 1.452, *p* < 0.0001)). After ∆^9^-THC administration (Figure 7B), the pulse distention did not significantly change with the highest dose tested (6 mg/kg). In contrast, the dose of 3 mg/kg slightly reduced pulse distension in the last 3 h of experiment (reduction of ~15%; significant effect of treatment (F_3, 2016_ = 51.44, *p* < 0.0001), time (F_71, 2016_ = 0.9891, *p* = 0.5045), and time x treatment interaction (F_213, 2016_ = 1.199, *p* = 0.0320)), and this effect was prevented by the treatment with AM 251 (6 mg/kg, i.p; Figure 7D; significant effect of treatment (F_3, 2016_ = 60.25, *p* < 0.0001), time (F_71, 2016_ = 0.7147, *p* = 0.9652), and time x treatment interaction (F_213, 2016_ = 0.8335, *p* = 0.9569)).

The administration of AM 630, which did not alter the pulse distension administered by itself, induced a reversion of JWH-018-induced effects, especially after one hour from injection (Figure 8A; significant effect of treatment (F_3, 2016_ = 81.55, *p* < 0.0001), time (F_71, 2016_ = 1.801, *p* < 0.0001), and time x treatment interaction (F_213, 2016_ = 1.223, *p* = 0.0200)), but it did not alter the ∆^9^-THC 3 mg/kg-induced effect (Figure 8B; significant effect of treatment (F_3, 2016_ = 50.25, *p* < 0.0001), time (F_71, 2016_ = 1.755, *p* < 0.0001), and time x treatment interaction (F_213, 2016_ = 1.027, *p* = 0.3842)).

The mixture of AM 630 and AM 251 administered by itself did not alter the pulse distension in mice (Figure 8C,D). The effect induced by JWH-018 was slightly increase after three hours from co-administration treatment (Figure 8C, significant effect of treatment (F_3, 2016_ = 89.90, *p* < 0.0001), time (F_71, 2016_ = 2.812, *p* < 0.0001), and time x treatment interaction (F_213, 2016_ = 0.9173, *p* = 0.7909)), while the ∆^9^-THC effect on pulse distension was further increase immediately after AM 630 and AM 251 injection (Figure 8D, significant effect of treatment (F_3, 2016_ = 30.53, *p* < 0.0001), time (F_71, 2016_ = 1.487, *p* = 0.0058), and time x treatment interaction (F_213, 2016_ = 1.476, *p* < 0.0001)).

### 2.4. Blood Pressure

Basal systolic and diastolic pressure (103 ± 5 mmHg, Figure 9 and Figure 10) did not change in vehicle-treated mice over the 6 h observation.

JWH-018 dose-dependently increased the basal systolic pressure (Figure 9A; significant effect of treatment (F_3, 1960_ = 196.2, *p* < 0.0001), time (F_69, 1960_ = 3.305, *p* < 0.0001), and time x treatment interaction (F_207, 1960_ = 1.161, *p* = 0.0607)). In particular, the highest dose (6 mg/kg, i.p) showed a systolic pressure increase of ∼30%, compared to the basal, which started from 20 min after injection and persisted until the end of the experiment. This effect was completely prevented by the pre-treatment with AM 251 (6 mg/kg, i.p; Figure 9B; significant effect of treatment (F_3, 1960_ = 409.5 *p* < 0.0001), time (F_69, 1960_ = 2.723, *p* < 0.0001), and time x treatment interaction (F_207, 1960_ = 1.728, *p* < 0.0001)). Basal diastolic pressure (74 ± 5 mmHg, Figure 7C) transiently increased only after JWH-018 6 mg/kg injection (increasing of ∼30% during the first 20 min, significant effect of treatment (F_3, 1960_ = 40.48, *p* < 0.0001), time (F_69, 1960_ = 1.017, *p* = 0.4392), and time x treatment interaction (F_207, 1960_ = 1.471, *p* < 0.0001)). This effect was completely prevented by the pre-treatment with AM 251 (6 mg/kg, i.p.; Figure 9D; significant effect of treatment (F_3, 1960_ = 45.85, *p* < 0.0001), time (F_69, 1960_ = 2.656, *p* < 0.0001), and time x treatment interaction (F_207, 1960_ = 1.414, *p* = 0.0002).

During the central hours of experiment, the hypertensive effect induced by JWH-018 was completely reverted following the pre-treatment of AM 630, inducing hypotension on both systolic (Figure 11A, significant effect of treatment (F_3, 1960_ = 302.2, *p* < 0.0001), time (F_69, 1960_ = 1.439, *p* = 0.0114), and time x treatment interaction (F_207, 1960_ = 2.533, *p* < 0.0001)) and diastolic (Figure 11C, significant effect of treatment (F_3, 1960_ = 199.6, *p* < 0.0001), time (F_69, 1960_ = 1.094, *p* = 0.2804), and time x treatment interaction (F_207, 1960_ = 2.280, *p* < 0.0001)), even if the AM 630 did not significantly differ to the vehicle.

Even the mixture of AM 630 and AM 251 did not alter the systolic and diastolic pressure administered by itself, but it was able to reduce the effect of JWH-018 on systolic pressure (Figure 11B, significant effect of treatment (F_3, 1960_ = 472.5, *p* < 0.0001), time (F_69, 1960_ = 0.7020, *p* = 0.9699), and time x treatment interaction (F_207, 1960_ = 1.845, *p* < 0.0001)), inducing a slight hypotension and diastolic pressure (Figure 11D, significant effect of treatment (F_3, 1960_ = 68.80, *p* < 0.0001), time (F_69, 1960_ = 0.6315, *p* = 0.9922), and time x treatment interaction (F_207, 1960_ = 0.6781, *p* = 0.9998)).

After ∆^9^-THC administration, systolic blood pressure (Figure 10A) significantly changed only at 35 and 50 min after 0.3 mg/kg injection and at 45 and 70 min after 3 mg/kg, while the diastolic pressure did not change (Figure 10B); significant effect of treatment (F_3, 1960_ = 39.27, *p* < 0.0001), time (F_69, 1960_ = 3.478, *p* < 0.0001), and time x treatment interaction (F_207, 1960_ = 1.007, *p* = 0.4621) and (effect of treatment (F_3, 1960_ = 59.11, *p* < 0.0001), time (F_69, 1960_ = 12.03, *p* < 0.0001), and time x treatment interaction (F_207, 1960_ = 0.7239, *p* = 0.9985), respectively)).

### 2.5. Breath Rate

The systemic administration of JWH-018 and ∆^9^-THC affected the breath rate in the mice (Figure 12).

Basal breath rate activity (243 ± 2 brpm) did not change in vehicle-treated mice over the 6 h observation. Basal breath rate decreased after highest doses JWH-018 administration (Figure 12A; significant effect of treatment (F_3, 1440_ = 229.9, *p* < 0.0001), time (F_71, 1440_ = 8.220, *p* < 0.0001), and time x treatment interaction (F_213, 1440_ = 2.186, *p* < 0.0001)). At 6 mg/kg dose, the breath rate was reduced by ∼65% at 10 min from the injection, up to 4 h from the injection (∼31% of decrease at 300 min), and the intermediate dose (3 mg/kg) reduced the breath rate ∼60% after one hour from injection. The ∆^9^-THC (Figure 12B) administration effects on breath rate were significant at 6 mg/kg dose, in which the breath rate decreased of ∼40% after 2 h from the injection until the end of the experiment (significant effect of treatment (F_3, 2016_ = 378.3, *p* < 0.0001), time (F_71, 2016_ = 8.218, *p* < 0.0001), and time x treatment interaction (F_213, 2016_ = 3.620, *p* < 0.0001). Treatment with AM 251 (6 mg/kg, i.p.) did not alter the heart rate by itself, but totally prevented effects on the breath rate induced by JWH-018 (6 mg/kg, Figure 12C; significant effect of treatment (F_3, 1728_ = 735.5, *p* < 0.0001), time (F_71, 1728_ = 16.25, *p* < 0.0001), and time x treatment interaction (F_213, 1728_ = 8.965, *p* < 0.0001)) and ∆^9^-THC (6 mg/kg, Figure 12D; significant effect of treatment (F_3, 2016_ = 321.2, *p* < 0.0001), time (F_71, 2016_ = 2.457, *p* < 0.0001), and time x treatment interaction (F_213, 2016_ = 2.832, *p* < 0.0001)).

The AM 630 administration did not differ from vehicle but, during the last two hours of experiment, however, it reverted the reduction of breath rate induced by both JWH-018 (Figure 13A, significant effect of treatment (F_3, 2016_ = 454.0, *p* < 0.0001), time (F_71, 2016_ = 11.13, *p* < 0.0001), and time x treatment interaction (F_213, 2016_ = 3.462, *p* < 0.0001)) and ∆^9^-THC (Figure 13B; significant effect of treatment (F_3, 2016_ = 235.2, *p* < 0.0001), time (F_71, 2016_ = 7.487, *p* < 0.0001), and time x treatment interaction (F_213, 2016_ = 2.670, *p* < 0.0001)).

The mixture of AM 630 and AM 251, did not alter the breath rate administered by itself, but it slightly increased the breath rate reduction induced by JWH-018 during the first two hours after antagonist’s injection (Figure 13C; significant effect of treatment (F_3, 2016_ = 623.4, *p* < 0.0001), time (F_71, 2016_ = 9.663, *p* < 0.0001), and time x treatment interaction (F_213, 2016_ = 4.258, *p* < 0.0001)). On the contrary, the effect induced by ∆^9^-THC was reverted by the co-administration during the last two hours of experiment (Figure 13D; significant effect of treatment (F_3, 2016_ = 222.1, *p* < 0.0001), time (F_71, 2016_ = 3.579, *p* < 0.0001), and time x treatment interaction (F_213, 2016_ = 2.967, *p* < 0.0001)).

### 2.6. Plethysmography

Respiratory parameters have been evaluated after JWH-018 and ∆^9^-THC 6 mg/kg administration (Table 2), according to dose–response curves (Figure 12A,B).

The vehicle injected during JWH-018 experiment induced an initial decrease of breathing frequency (Figure 12A), and this value was monitored with plethysmography analysis (Table 2, a’), which reported an fr_ie decrease of ∼12% (significant effect of treatment t = 34.63, df = 28, *p* < 0.001), with a consequent BB_ie increase of ∼14% (significant effect of treatment t = 30.91, df = 28, *p* < 0.001), whereas the electrical parameters that represent the last hours of experiment did not change (Table 2, b’). On the other hand, concerning the ∆^9^-THC experiment, the vehicle injection did not change basal values (Figure 9B), and this is also demonstrated in plethysmography electrical parameters (Table 2).

The JWH-018 6 mg/kg induced a deep reduction of breath rate during the first 2 h after injection, and the plethysmography electrical values referred to this reduction were evaluated (Table 2). In particular, JWH-018, when induced with a decrease of fr_ie of ~50% (significant effect of treatment t = 72.78, df = 28, *p* < 0.001), also induced an increase of BB_ie of ~78% (significant effect of treatment t = 79.76, df = 28, *p* < 0.001), a decrease of TV of ~16% (significant effect of treatment t = 24.62, df = 28, *p* < 0.001), and an increase of RT of ~60% (significant effect of treatment t = 24.39, df = 28, *p* < 0.001). During the last hour of experiment, JWH-018-treated mice slightly recovered breathing function (Figure 12A). Despite this, plethysmography electrical parameters related to this recovery (Table 2) showed a reduction of fr_ie than basal of ∼30% (significant effect of treatment t = 49.39, df = 28, *p* < 0.001) and a consequent BB_ie increase of ~45% (significant effect of treatment t = 54.28, df = 28, *p* < 0.001), together with a TV decrease (~8%, significant effect of treatment t = 6.759, df = 28, *p* < 0.001) and RT increase (~50%, significant effect of treatment t = 27.13, df = 28, *p* < 0.001). After ∆^9^-THC 6 mg/kg administration, dose–response curve initially showed a slight, but not significant, decrease of breath rate (Figure 12B). Referred to this, the analysis of plethysmography parameters (Table 2), showed a decrease of ~15% of fr_ie, compared to basal (significant effect of treatment t = 40.07, df = 28, *p* < 0.001), and an increase of ~15% of BB_ie (significant effect of treatment t = 30.27, df = 28, *p* < 0.001), together with an increase of ~20% of RT (significant effect of treatment t = 19.43, df = 28, *p* < 0.001). During the last hours of injection ∆^9^-THC induced a deeper decrease of breath rate (Figure 12B), and the plethysmography parameters were analyzed (Table 2), which showed a fr_ie values decrease (~30%, significant effect of treatment t = 89.74, df = 28, *p* < 0.001), a BB_ie increase (~40%, significant effect of treatment t = 68.28, df = 28, *p* < 0.001), a TV decrease (~6%, significant effect of treatment t = 3.283, df = 28, *p* = 0.0028), and a RT increase (~40%, significant effect of treatment t = 625.37, df = 28, *p* < 0.001), with respect to the basal values.

### 2.7. Oxygen Saturation

The oxygen saturation rate (99.1 ± 1.2 SpO_2_) did not change in the vehicle-treated mice over the 6 h observation. The basal oxygen saturation (98.9 ± 1.3% SpO_2_; Figure 14A) was transiently decreased by JWH-018 at 6 mg/kg, with a reduction of ∼17% at 35 min from injection (significant effect of treatment (F_3, 1440_ = 56.27, *p* < 0.0001), time (F_71, 1440_ = 4.881, *p* < 0.0001), and time x treatment interaction (F_213, 1440_ = 3.501, *p* < 0.0001)), and the effect was totally prevented by the treatment with AM 251 (6 mg/kg, i.p; Figure 14B; significant effect of treatment (F_3, 1728_ = 64.54, *p* < 0.0001), time (F_71, 1728_ = 9.732, *p* < 0.0001), and time x treatment interaction (F_213, 1728_ = 3.682, *p* < 0.0001)) and by the treatment with AM 630 (6 mg/kg, i.p; Figure 14C; significant effect of treatment (F_3, 2016_ = 36.42, *p* < 0.0001), time (F_71, 2016_ = 4.699 *p* < 0.0001), and time x treatment interaction (F_213,2016_ = 1.819, *p* < 0.0001)).

The co-administration (Figure 14D) of the mixture of antagonists (AM 630 + AM 251) did not significantly revert the reduction of oxygen saturation induced by JWH-018.

Differently, ∆^9^-THC administration did not significantly affect the arterial oxygen saturation.

## 3. Discussion

The present study demonstrates that JWH-018 administration deeply affected the cardiovascular and respiratory functions in awake mice. Specifically, JWH-018 altered the cardiac responses by triggering long-lasting bradycardia, and bradyarrhythmia interspersed with sudden episodes of tachyarrhythmias, characterized by prolonged PR interval, increase in QRS complex duration, and changes in QT and QTcF interval. JWH-018 also provoked vasoconstriction and a long-lasting increase of systolic pressure, paralleled by a transient rise of the diastolic one. Moreover, the prolonged bradypnea, characterized by TV reduction, RT augment, and SpO_2_ decrease, was measured after JWH-018 exposure. Cardiovascular and respiratory alterations caused by JWH-018 were significantly more potent than those determined after Δ^9^-THC administration, which induced (i) bradycardia and bradypnea at the highest tested dose (6 mg/kg) only, (ii) a slight and transient tachycardia at the lowest dose (0.3 mg/kg) and (iii) a slight vasoconstriction at the intermediate dose (3 mg/kg). All these adverse effects were immediately and totally prevented by AM 251 administration, demonstrating their dependency on CB_1_ receptors activation. Despite this, the treatment with AM 630 and with the mixture of AM 251 and AM 630 also modulated the effect JWH-018- and Δ^9^-THC-induced bradycardia, suggesting a possible mechanism that could also be mediated by CB_2_ receptors. In particular, the AM 630 treatment totally reverted the JWH-018-induced bradycardia and slightly improved the Δ^9^-THC-induced bradycardia, and the effect was more delayed than that induced by AM 251. The administration of the mixture of both CB antagonists prevented the bradycardic effects induced by both JWH-018 and Δ^9^-THC, as well as with a delayed action, when compared to AM 251-antagonism, but it was more rapid than AM 630-antagonism. Vascular parameter alterations (both pulse distension, systolic and diastolic pressure) induced by JWH-018 were totally reverted with AM 630, in particular the CB_2_ antagonist treatment induced an opposite effect. The co-administration of AM 251 and AM 630 did not have any effect on pulse distension, but it reverted both diastolic and systolic pressure. Bradypnea and SpO_2_ reduction induced by JWH-018 was reverted by AM 630 administration, and only bradypnea was partially reverted after the co-administration of CBs antagonist. On the contrary, the effect induced by Δ^9^-THC on breath rate was partially reverted by AM 630 alone, and it was totally reverted by AM 251 and AM 630 co-administration. Our current findings are in line with the literature reporting the signs and symptoms of several hospitalized intoxicated patients [8,12,15,22,45,46,47], clearly pointing out the severity of cardiovascular and respiratory consequences that JWH-018 and SCs users may face.

### 3.1. Cardiovascular Effects

Cannabinoids could affect cardiovascular functions through the stimulation of CB receptors expressed both at the peripheral and central levels [29,48,49,50]. In particular, CB receptor activation may control the cardiovascular activity by modulating the prejunctional inhibition of transmitter release from postganglionic sympathetic neurons, by central sympathoexcitation, through vagal activation at the brain stem level, and also by central sympathoinhibition [29,50].

The currently described bradycardic effect caused by both JWH-018 (at all doses tested) and Δ^9^-THC (at the highest dose only) is consistent with the data obtained in previous preclinical studies using cannabis and other SCs [26,28,50,51,52,53,54,55]. Bradycardia is mainly ascribable to the increase of vagal activity and reduction of sympathetic tone, as suggested by former in vivo investigations using Δ^9^-THC doses ranging from 1 mg/kg to 5 mg/kg [50,51,56]. Conversely, lower doses of Δ^9^-THC have been demonstrated to induce a mild tachycardic effect, due to sympathetic stimulation [57,58].

Concerning the central nervous system (CNS), one of the main brain areas involved in the modulation of cardiovascular function is the medulla oblongata, specifically the medullary cardiovascular center, in which the sympathetic nervous system (SNS) and peripheral nervous system (PNS) outflows are coordinated [53,59,60]. In particular, bradycardia could be directly caused by CB_1_ receptor activation into the cerebellomedullary cistern, which could provoke the direct enhancement of vagal activity [50]. In fact, previous studies reported that Δ^9^-THC administration in rodents has been able to inhibit sympathetic activity, thus increasing the parasympathetic tone, leading to bradycardia [51,61]. Enhancement of vagal tone and CB_1_-mediated sympathetic inhibition were also reported, describing a bradycardic effect induced by different SCs, e.g., WIN55212-2 and CP55940 [27].

Nevertheless, CB_1_ receptors stimulation into the medulla also leads to sympathetic pathways activation, consequentially provoking noradrenaline release in many sympathetically innervated tissues, such as the cardiac ones [50,53]. Indeed, beyond their central action, SCs can affect peripheral autonomic neurons [35]. SCs could pre-synaptically inhibit the release of noradrenaline from many postganglionic sympathetic neurons, provoking the decrease of heart rate, in accordance with previous in vivo and in vitro studies [50,53,62,63,64]. This mechanism could also be due to CB_1_-mediated calcium channel inhibition [64,65].

Recently, it has been reported that vagal activity is chiefly implicated in the cardiotoxic effects observed in human intoxications [35]. The recent evidence is in line with previous clinical data describing symptoms such as bradyarrhythmia, asystole, atrioventricular block, and cardiac arrest after cannabis and SCs assumption [15,66,67,68,69,70].

Noteworthy, the CB_2_ receptor selective antagonist AM 630 also improved the bradycardia induced by JWH-018 and Δ^9^-THC, indicating a CB_2_ receptor involvement in this effect. In particular, AM 630 elicited its effect later than AM 251, suggesting a primary central effect on CB_1_ and then a peripheral effect on both CB_1_ and CB_2_. Nevertheless, Krylatov and colleagues study showed that the CB_2_ receptor antagonist SR144528 does not have a direct effect on heart rate, but it can modulate the cAMP [71]. The cAMP is responsible for several actions, including the modulation of the heart rate through catecholaminergic control [72]. Even the improvement of heart rate, following the co-administration of both CB_1_ and CB_2_ antagonist’s treatment, suggests the involvement of both receptors, which are present in the myocardium of mice [73]. Despite this, the co-administration of the two antagonists showed a more rapid effective action in reverting the bradycardia induced by JWH-018 and Δ^9^-THC, compared with the administration of AM 630 alone, probably due to the prevailing action on CB_1_.

It is well-known that the vagal activation should induce both bradycardia and hypotension [35,60,74]. This latter outcome was not evidenced in our current investigation. Nonetheless, our present data are in accordance with previous studies reporting that the synthetic cannabinoid WIN 55212-2 induced both a deep bradycardia and a blood arterial pressure increase on conscious normotensive rats [75].

Our findings suggest that both the sympathetic and parasympathetic systems could contribute to the effects induced by both JWH-018 and Δ^9^-THC. The increase of sympathetic activity could explain the hypertension induced by JWH-018 and the tachyarrhythmias events provoked by both cannabinoids [31,60,76]. Centrally, the CB_1_ stimulation into dorsal periaqueductal gray (dPAG) could explain the sympathetic enhancement and the consequent blood pressure increase [60]. Moreover, the activation of the CB_1_ receptor in the perivascular sympathetic nerve leads to neurotransmitters modulation, such as γ-aminobutyric acid or glutamate and other monoamines, which causes a vasoconstrictor effect and hypertension [35,77,78]. These blood pressure effects were reverted by AM 251, but also AM 630, despite the fact that there is no evidence regarding the involvement of CB_2_ receptor in vascular regulation [79,80]. Specifically, our finding showed a hypotensive effect after the pretreatment of AM 630, which is inconsistent with a preclinical study in mice [81]. In fact, further studies would be needed to evaluate the effect of AM 630 or other CB_2_ selective receptor antagonists, in order to understand how CB_2_ receptors could be involved in a hypotensive effect or if other receptors (as TRPV or GPR55 receptors) [35,79,80] could be involved. Although this is an unexpected result, the co-administration of both AM 251 and AM 630 reduced the hypotensive effect, again suggesting a prevalent sympathetic effect mediated by CB_1_.

The stimulation of sympathetic fibers could also provoke inotropic, chronotropic, and dromotropic positive effects, which may increase the risk of tachyarrhythmias [35]. Electrophysiological studies showed that Δ^9^-THC was able to enhance sinus node automaticity and sinus-atrial and atrio-ventricular conduction favoring arrhythmias insurgence [58,82]. The wide QRS complex and PR and QT interval prolongation found after JWH-018 and Δ^9^-THC administration confirm the presence of an irregular cardiac rhythm.

The QRS complex enlargement has been shown in patients affected by Brugada syndrome, an inherited arrhythmic disease pertinent to channelopathies [83], but also in the development of a Δ^9^-THC-induced Type I Brugada pattern at ECG in a 36-year-old healthy man [84]. Moreover, a prolonged QRS complex is characteristic of Osborne wave [85], which is an arrhythmic index and was detected in hypothermic SCs users [86]. This evidence suggests that hypothermia induced by JWH-018 [87], as well as by other SCs [28,88,89], could enhance arrhythmic events. Even the QT interval prolongation was detected after JWH-018 administration, and this result is consistent with Yun and colleagues’ studies, which found the same evidence with JWH-030 treatment [90]. In particular, QT interval prolongation was better linked to tachyarrhythmia, in which our results showed both prolonged QT and prolonged QTcF. The QT interval represents ventricular repolarization, and its prolongation could depend on the rectifier potassium channel, Ikr, which is encoded by human Ether-à-go-go-related (hERG) gene [91,92]. The JWH-018 could prolong QT interval whit blockage of hERG channel, as shown in a JWH-030 study, which inhibited hERG with affinity of 88.36 μM [90]. Torsade de pointes (TdP) is the principal type of arrhythmia associated with delayed ventricular repolarization, indicated by QT interval prolongation [93]. TdP was reported in a patient who abused K2 products [94]. Moreover, JWH-018 during the tachyarrhythmia period and, to a lesser extent, Δ^9^-THC also induced the PR interval prolongation that is attributable to a higher risk of atrial fibrillation [95], and this evidence was reported in a clinical case after SCs consumption [96].

Although the relationship between sympathetic over activity and the development of arrhythmias is well-known, vagal activation could also play a key role [97,98,99,100,101]. Both the Ca^2+^ intracellular-release, sympathetic-mediated, and potential action duration reduction vagal-induced could contribute to the triggered activity, due to an increase of Na^+^/Ca^2+^ exchanger current [102,103,104]. Ion-channel dysregulation may result in conduction disturbances within the heart, and it could increase the risk of arrhythmias [35], due to the voltage-gated channels that are involved in each phase of cardiac potential action [105]. Arrhythmias can be induced not only by CB-dependent mechanisms, but also by acting on other targets in a CB-independent manner [34,35]. Similarly, Δ^9^-THC-induced tachycardia is related to biphasic and dose-dependent effects, and low or moderate doses cause a direct sympathetic stimulation, leading to tachycardia, hypertension, and increased cardiac output, and a higher dose a Δ^9^-THC-induced parasympathetic vasodilatation with bradycardia and hypotension [57,106]. The autonomic dysregulation results in increased cardiac output and workload, thus increasing the myocardial oxygen demand induced by Δ^9^-THC, coupled with microvascular/coronary artery spasm and the prothrombotic state arising from Δ^9^-THC use, which has been associated with an increased risk of acute coronary syndrome, even in the absence of an atherosclerotic coronary artery disease [107]. Consequently, the use of cannabis should be cautioned in patients with a high cardiovascular risk profile or affected by cardiovascular diseases and avoided 2 h before physical exercise [108].

In fact, in vitro and in vitro ex vivo studies on rodents demonstrated the inhibition of subtypes of calcium, sodium, and potassium voltage-gated cardiac channels endogenous cannabinoid-provoked [109,110]. Even non-cannabinoids receptors, such as GPR55 [111,112], peroxisome proliferator-activated receptor (PPAR), TRP channels, and β-adrenoceptors could modulate ion cardiac channel and alter cardiac regulations, thus leading to cardiac arrhythmias [34,113,114,115]. This evidence could suggest the involvement of non-cannabinoids receptors in arrhythmic response modulation. Despite this, the action of the CB_1_ selective antagonist, AM 251, or CB_2_ selective antagonist, AM 630, and their co-administration reverted JWH-018 and Δ^9^-THC adverse effects, attributing these latter to the cannabinoid receptors stimulation, but higher doses than those tested the stimulation of non-cannabinoids receptors should not be excluded.

### 3.2. Respiratory Effects

Beyond cardiovascular effects, both JWH-018 (all doses) and Δ^9^-THC (the highest dose) also affected the respiratory activity. A reduction of breath rate, combined with a transient decrease of SpO_2_ induced by JWH-018, is in line with the other studies carried out on rodents [27,28,116,117].

Even the respiratory function could be impaired because of central or peripheral action [27]. Indeed, cannabinoids could bind CB_1_ receptors into several brainstem portions [118]. The decrease of breathing frequency could start by central vagal alteration, followed by peripheral receptors involvement [119,120]. The peripheral action on cannabinoids has been reported in previous studies [120]. The peripheral bind with CB_1_ receptor found on an axon terminals of airways nerves or with other receptors, such as chemoreceptors, baroreceptors, and pulmonary stretch receptors, can increase bronchial airway resistance, explaining a possible mechanism of cannabinoid-induced respiratory depression [27,39,120]. Beyond these mechanisms, even an irregular rhythmogenic activity could affect the breathing frequency [120], and these, together with the previous hypothesis, could also explain the reduction of TV and the increase of RT.

Moreover, the mitochondrial CB_1_ receptors in striated muscle that participate in the oxygen regulation could explain the reduction of SpO_2_ [121]. All evidence has been confirmed by clinical data [23].

Beyond CB_1_-mediated action, the CB_2_ receptor involvement on JWH-018- and Δ^9^-THC-induced bradypnea and SpO_2_ reduction should not be excluded. Indeed, the administration of AM 630 was able to revert these effects. Specifically, CB_2_ receptors are expressed on lung tissue, mainly in the fibroblast and pulmonary macrophage [37]. Moreover, the CB receptors, in particular CB_2_, are mostly expressed in the immune cells of the respiratory system (i.e., eosinophil, monocyte, dendritic cell, mast cell, macrophage, B cell, or T cell) [122]. Although the activation of these cells should cause a protection of airway [123], the CB_2_ stimulation could worsen some respiratory parameters, such as the airway hyper-responsiveness and the eosinophil influx into the airway, as shown for JWH-133, a CB_2_ agonist [124]. This could suggest how the binding of the CB_2_ receptor could affect the breath rate. Despite this, the obtained evidence that the block of both CB_1_ and CB_2_ did not completely revert the bradypnea and SpO_2_ reduction of JWH-018 suggests that other non-cannabinoid mechanisms could be involved.

## 4. Materials and Methods

### 4.1. Animals

A total of 304 male outbred ICR (CD-1^®^) mice weighing 30–35 g (Centralized Preclinical Research Laboratory, University of Ferrara, Italy) were group-housed (5 mice per cage; floor area per animal was 80 cm^2^; minimum enclosure height was 12 cm), exposed to a 12:12-h light-dark cycle (light period from 6:30 AM to 6:30 PM) at a temperature of 20–22 °C and humidity of 45–55%, and provided ad libitum access to food (Diet 4RF25 GLP; Mucedola, Settimo Milanese, Milan, Italy) and water during the entire time the animals spent in their home cages. The experiments were performed during the light phase. The experimental procedures were approved by the Italian Ministry of Health (license n° 223/2021-PR and extension CBCC2.46.EXT.21), by the Animal Welfare Body of the University of Ferrara and were performed in accordance with the U.K. Animals (Scientific Procedures) Act of 1986 and associated guidelines, and by the new European Communities Council Directive of September 2010 (2010/63/EU). According to the ARRIVE guidelines, all efforts were made to minimize animals’ pain and discomfort, and to reduce the number of experimental subjects. For the overall study, a total of 176 mice were used and divided into different groups, as follows. Cardiorespiratory studies—JWH-018 treatment (vehicle or 3 different doses, i.e., 0.3, 3, and 6 mg/kg): a total of 24 mice were used (6 mice/condition); cardiorespiratory studies—∆^9^-THC treatment (vehicle or 3 different ∆^9^-THC doses, i.e., 0.3, 3, and 6 mg/kg): a total of 32 animals were employed (8 mice/treatment); cardiorespiratory experiments—JWH-018 or ∆^9^-THC combined with AM 251 (6 mg/kg): 24 mice were utilized (8 mice for each treatment, namely AM 251 alone, JWH-018 + AM 251 or ∆^9^-THC + AM 251)—JWH-018 or ∆^9^-THC combined with AM 630: 24 mice were utilized (8 mice for each treatment, namely AM 630 alone (6 mg/kg), JWH-018 + AM 630 or ∆^9^-THC + AM 630)—JWH-018 or ∆^9^-THC combined with the co-administration of AM 251 (6 mg/kg) and AM 630 (6 mg/kg): 24 mice were utilized (8 mice for each treatment, namely AM 251 + AM 630 alone, JWH-018 + AM 251 and AM 630 or ∆^9^-THC + AM 251 and AM 630). Electrocardiograms and plethysmography analyses—JWH-018 treatment (vehicle or JWH-018 6 mg/kg): 8 mice were employed in total (4 mice/treatment); electrocardiograms and plethysmography analyses—∆^9^-THC (vehicle or ∆^9^-THC 6 mg/kg): a total of 8 mice were used (4 mice/condition). Blood pressure studies: (i) JWH-018 treatment (vehicle or 3 different doses, i.e., 0.3, 3, and 6 mg/kg, or AM 251 alone or JWH-018 + AM 251): 48 mice were employed in total (8 mice/treatment); (ii) ∆^9^-THC treatment (vehicle or 3 different ∆^9^-THC doses, i.e., 0.3, 3, and 6 mg/kg or AM 251 alone or ∆^9^-THC + AM 251): a total of 32 mice were utilized (8 animals/treatment); (iii) JWH-018 combined with AM 630 or co-administration of AM 251 (6 mg/kg) and AM 630 (6 mg/kg): 40 mice were utilized (8 mice for each treatment, vehicle, AM 630 alone, AM 251 + AM 630, JWH-018 + AM 630 or JWH-018 + AM 251 and AM 630); (iv) ∆^9^-THC combined with AM 630 or co-administration of AM 251 (6 mg/kg) and AM 630 (6 mg/kg): 40 mice were utilized (8 mice for each treatment, vehicle, AM 630 alone, AM 251 + AM 630, ∆^9^-THC + AM 630, or ∆^9^-THC + AM 251 and AM 630)

### 4.2. Drug Preparation and Dose Selection

JWH-018 and ∆^9^-THC were purchased from LGC Standards (LGC Standards S.r.L., Sesto San Giovanni, Milan, Italy), while AM 251 and AM 630 was obtained from Tocris (Tocris, Bristol, UK). Drugs were initially dissolved in absolute ethanol (final concentration was 5%) and Tween 80 (2%) and brought to the final volume with saline (0.9% NaCl). The solution made with ethanol, Tween 80, and saline were also used as the vehicle. The CB_1_ receptor-preferring antagonist/inverse agonist AM 251 (6 mg/kg) or CB_2_ receptor-selective antagonist AM 630 (6 mg/kg) or the co-administration of AM 251 and AM 630 were administered 60 min after JWH-018 and ∆^9^-THC injection in cardiovascular experiments and 20 min before JWH-018 and ∆^9^-THC injections in blood pressure analyses. Drugs were administered by intraperitoneal injection (i.p.) in a volume of 4 µL/g.

### 4.3. Evaluation of Cardiorespiratory and Blood Pressure Changes

To monitor the cardiorespiratory parameters in awake and freely moving animals, without using invasive instruments and handling, a collar with a sensor was used to detect heart rate, breath rate, oxygen saturation, and pulse distension with an acquisition frequency of 15 Hz [28,125,126,127,128]. During experiment, mice were allowed to freely move around their cages (30 × 30 × 20 cm), while having no access to food or water and while being monitored by the sensor collar through the software MouseOx Plus (STARR Life Sciences^®^ Corp, Oakmont, PA, USA). In the first hour of acclimation, a fake collar, similar in design to the collar used in the test, but without a sensor, was used to minimize the potential stress of animals during experiment. Then, the collar with the sensor was applied, while the baseline parameters were monitored for 60 min. Subsequently, drugs or the vehicle were administered, and data were recorded for 5 h.

ECG and plethysmography parameters were collected from conscious animals using a non-invasive ECG and plethysmography TUNNEL system with an acquisition frequency of 1000 Hz (Emka Technologies, Paris, France). All ECG recording sessions were performed during daytime, and data were analysed using the *iox2* data acquisition analysis software (Emka Technologies). Each mouse was put inside the tunnel, which was then closed, ensuring the animal was properly restrained. To minimize the effects of stress, animals were allowed to stay in the restraining system for 1 min before starting recordings. Indeed, direct observation of the animals, ECG, and plethysmography traces proved that they were calm and that heart rate and breath were stable. The experiment provides baseline recording lasting for 15 min, followed by a recording session of 45 min during/after vehicle or drugs exposure. For data acquisition, a series of repeated measurements were performed on the same animal at each time point, and data for the same animal were collected over the different week intervals [129,130].

As previously reported [125], systolic and diastolic blood pressure were measured by tail-cuff plethysmography using a BP-2000 blood pressure analysis system (Visitech Systems, Apex, NC, USA). For each session, mice were placed in a metal box restraint with its tail passing through the optical sensor and compression cuff before finally being taped to the platform. A traditional tail-cuff occluder was placed proximally on the animal’s tail, which was then immobilized with tape in a V-shaped block between a light source above and a photoresistor below. Upon inflation, the occluder stopped blood flow through the tail, while upon deflation, the sensor detected the blood flow return. The restraint platform was maintained at 37 °C. Before experiments, mice were acclimated to restraint and tail-cuff inflation for 5–7 days. On the test day, 10 measurements were made to collect basal blood pressure. Upon the tenth analysis, the software was paused, and mice were injected with either drug treatments or the vehicle; animals were then repositioned in the restraints, and 60 measurements were acquired.

### 4.4. Data and Statistical Analysis

Data related to heart rate (heart beats per min; bpm), pulse distention (vessel diameter changes; μm), breath rate (respiratory rate per minute; rrpm), and SpO_2_ saturation (oxygen blood saturation) were expressed as % changes of basal values. Tachyarrhythmia analysis expressed in histograms represents the number of tachyarrhythmia events registered each hour for 6 consecutive hours. Abnormal heart rhythms were considered tachyarrhythmia events when heartbeat, after vehicle or diverse drug administration, was almost 200 pulses higher, compared to the average of the basal rate. The statistical analysis of dose–response curve and tachyarrhythmia frequencies after exposure to different substances were performed by two-way ANOVA, followed by Bonferroni’s test for multiple comparisons.

Regarding ECG and plethysmography, Emka ECG and plethysmography analyzer software were used to analyze tracings recorded during data acquisition. ECG measurements were recorded in two experimental phases (before and after treatment, based on Mouse Ox’ results of heart rate and breath rate values). Based on mouse Ox profile, during EMKA analysis, a 1-min time window was chosen, evaluating recorded parameters with an acquisition frequency of 1000 Hz. For the ECG analysis, HR (bpm; heart rate), RR (ms; time between two consecutive peaks), PR (ms; time from the onset of the P wave to the start of the QRS complex), QRS (ms; is the time from the start to the end of the QRS complex), and QT (ms; time from the beginning of the QRS complex to the end of the T wave) parameters were determined. To better determine the relationship between the HR and the duration of the ventricular electrocardiogram (QT interval), as well as QTcF, were determined. To calculate the latter parameter, the Fridericia formula (QTcF = QT/∛ RR) was used [131,132]. Moreover, ECG tracks and single ECG waves were exported, including PR interval, QRS complex, and QT interval measurements.

For plethysmography, FR_IE (bpm; breathing frequency), BB_IE (ms; breath length computed by adding inspiration and expiration duration), TV (mL/s; tidal inspiration volume); EXP/INSP_V_RATIO__AVER (ratio of expired volume/tidal volume), and RT (ms; relaxion time) were determined. All parameters were analyzed using the *iox2* analyze software.

Changes in systolic and diastolic blood pressure were expressed as absolute values (mmHg). The effects of different concentrations of each substance over time were analyzed by a two-way ANOVA, followed by Bonferroni’s test for multiple comparisons where appropriate. The significance level was set at *p* < 0.05. Data were reported as mean standard error of the mean (SEM) of at least 6 independent experiments. Results were expressed as follows: (i) JWH-018 and ∆^9^-THC cardiovascular data: percentage change of baseline; (ii) JWH-018 and ∆^9^-THC tachyarrhythmia data: number of events; concerning blood pressure experiments, data were expressed as absolute values (mm/Hg), with mean ± SEM of 8 independent experiments. ECG analyzed results were related to HR dose–response curve, showing electrical variations referred to HR variation of vehicle and treated mice. Plethysmography analyzed data were related to BR dose–response curve and displayed electrical variations referred to BR variations of vehicle and treated mice. JWH-018 and ∆^9^-THC ECG and plethysmography recordings were analyzed using Student’s *t*-test for each basal and after-treating parameter comparisons, and data were expressed as mean overall effects of 15 evaluation for baseline and after treatment (mean ± SEM of 15 different evaluations). All statistical analyses were performed using GraphPad Prism software (GraphPad Prism, San Diego, CA, USA).

## 5. Conclusions

Our findings confirmed a deeper and more potent effect of JWH-018 than Δ^9^-THC on cardio-vascular and respiratory functions. This evidence is in line with the well-known action of JWH-018, which elicits its effects binding CB_1_ and CB_2_ receptors as a full agonist, in contrast to Δ^9^-THC, which is a partial agonist [47]. Specifically, all JWH-018- and Δ^9^-THC-induced effects can be attributed of both CB_1_ and CB_2_ receptors binding, since these outcomes were prevented by treatment with AM 251, a selective CB_1_ receptors antagonist, AM 630, a selective CB_2_ antagonist, and the co-administration of both. Despite this, the mechanism behind CB_2_ is still little known. Indeed, JWH-018 has a higher affinity for the CB_1_ and CB_2_ receptors than Δ^9^-THC (CB_1_ Ki of about 9.53 ± 0.88 nM, CB_2_ Ki of about 2.9 ± 2.7 nM for JWH-018, CB_1_ Ki of about 40.7 ± 1.7 nM, and CB_2_ Ki of about 36.4 ± 10 nM for Δ^9^-THC) [41,47,133].

Moreover, the lipophilic profile and different metabolism of two substances could also explain the different entities of effects [134,135]. The difference between the effects induced in animals’ in vivo experiment and data reported in clinical cases could be due to different factors, such as the route of administration, pharmacogenetic sensitivity, or dosage [35,48]. Indeed, low doses of exogenous cannabinoids drive sympathetic stimulation, while higher doses drive vagal stimulation [58,61,101]. In line with the human equivalent dose (HED) formula and with the dosage scale reported by users, in this work, the range of doses, from common to heavy, were tested [136,137]. The highest doses administered in mice were higher than the dosage labeled as “heavy” in the forums by users [136,137,138], and this could explain the greatest vagal response in mice. This evidence could also enlighten several intoxication cases reporting bradycardia as collateral symptoms, therefore highlighting the potential risk related to the consumption of toxic dosage.

In summary, the current findings confirmed the dangerous effect caused by SCBs on cardio-vascular and respiratory systems. In particular, JWH-018 induced bradycardia, bradyarrhythmia with sudden episodes of tachyarrhythmias, hypertension, and bradypnea, which could elicit its effects through several mechanisms CB-dependent that could involve the sympathetic and vagal systems.

It should also be underlined that, currently, any specific antidote for SCs intoxication is available. Antagonists of the CB_1_ receptor, which block the acute effects of cannabis employed for some years, were recently banned because of the psychiatric side effects, and there is no evidence concerning the use of CB_2_ antagonists as antidotes for SCs intoxication. Thus, at present, the clinical management is merely supportive and symptomatic. Various adverse effects associated with acute SCs intoxication are undistinguishable to some withdrawal signs and, therefore, similarly treated. Benzodiazepines are usually administered as a first-line treatment in patients experiencing irritability, agitation, anxiety, and seizures associated with intoxication or withdrawal. Neuroleptics and antipsychotic are also administered for severe or persistent psychosis and mania [139,140]. Transitory suicidal ideation may arise during acute intoxicated condition, and even worse, psychiatric symptoms may last longer than a week deserving rigorous evaluation for a comorbid psychiatric disorder, also based on the fact that many patients are polysubstance users [141,142]. Nonetheless, a broad consensus and clinical practice guidelines are still lacking regarding the patients’ management during SCs intoxication. Partial inhibition of the cardiovascular effects of Δ^9^-THC could be achieved with prior beta-blockers administration [143].

Therefore, to better clarify the mechanisms behind the cardiovascular damage of SCs, further studies will be needed, with the final aim at suggesting the possible treatments and therapeutic antidotal strategy enabling the physicians, which operates in the emergency departments to promptly and effectively manage the SCs in intoxicated patients. Together with the confirmed CB_1_-mediated mechanism beside cardiovascular effects, our current data could suggest the importance of CB_1_ antagonist (i.e., rimonabant) as an antidote in SCs-related acute intoxication cases. Moreover, further studies on CB_2_-mediated mechanisms are needed, in order to better understand how this receptor modulates the cardio-vascular and respiratory effects of SCs.

## Figures and Tables

**Figure 1 ijms-24-01631-f001:**
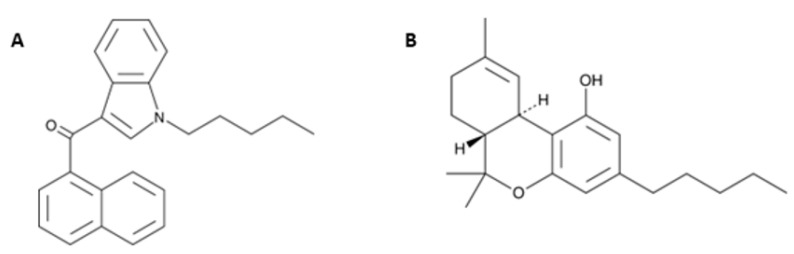
Chemical structures of JWH-018 (1-pentyl-3-(1-naphthoyl)indole; panel (**A**) and Δ^9^-THC (Δ^9^-Tetrahydrocannabinol; panel (**B**) from the Cayman Chemical website (https://www.caymanchem.com, last access on Tuesday 30 November 2021, at 3.30 pm).

**Figure 2 ijms-24-01631-f002:**
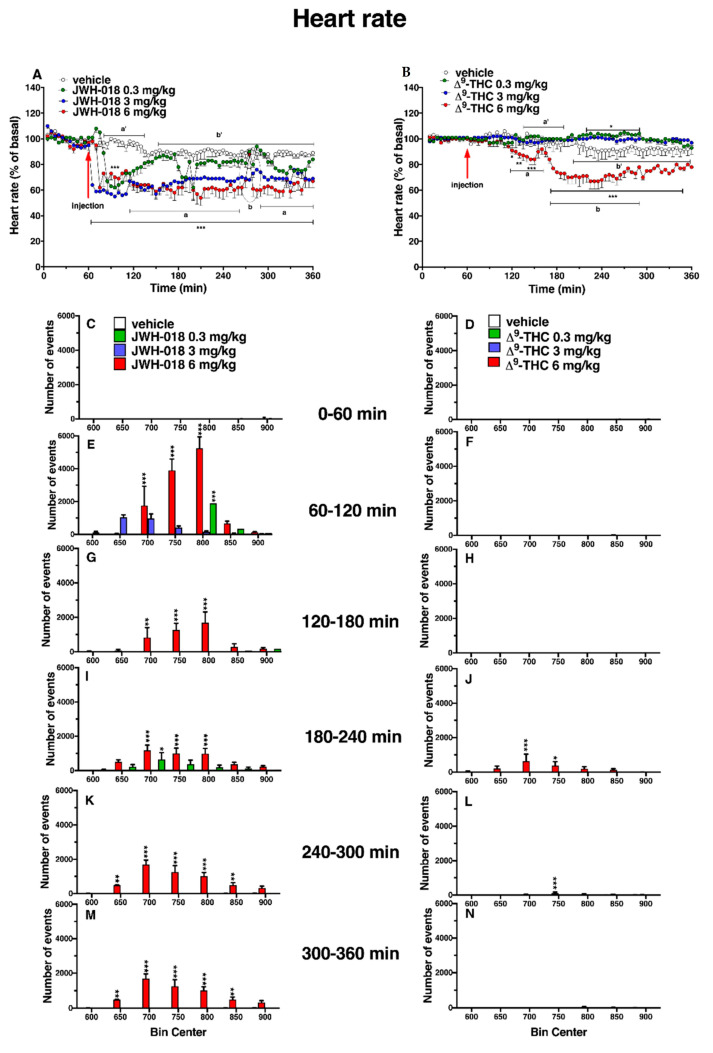
Effect of systemic administration of JWH-018 (0.3–6 mg/kg, panel (**A**), and ∆^9^-THC (0.3–6 mg/kg, panel (**B**)) on heart rate. Data are expressed as percentage of basal values in the form MEAN ± SEM of 6 different evaluations for JWH-018 experiments and MEAN ± SEM of 8 different evaluations for ∆^9^-THC experiments. Statistical analysis was performed by two-way ANOVA, followed by Bonferroni’s test for multiple comparisons. * *p* < 0.05 ** *p* < 0.01 *** *p* < 0.001 versus vehicle. Vehicle and 6 mg/kg dose curves are marked by letters (a’, b’ for vehicle and a, b 6 mg/kg treatment) to compare effects with ECG electrical parameters (Table 1). Frequency of tachyarrhythmia episodes after administration of JWH-018 (0.3–6 mg/kg, panels (**C**,**E**,**G**,**I**,**K**,**M**) and ∆^9^-THC (0.3–6 mg/kg, panels (**D**,**F**,**H**,**J**,**L**,**N**), expressed as number of events per mean heart rate value in the form MEAN ± SEM of 6 different evaluations for JWH-018 experiments and MEAN ± SEM of 8 different evaluations for ∆^9^-THC experiments. Statistical analysis was performed by two-way ANOVA, followed by Bonferroni’s test for multiple comparisons. * *p* < 0.05 ** *p* < 0.01 *** *p* < 0.001 versus vehicle.

**Figure 3 ijms-24-01631-f003:**
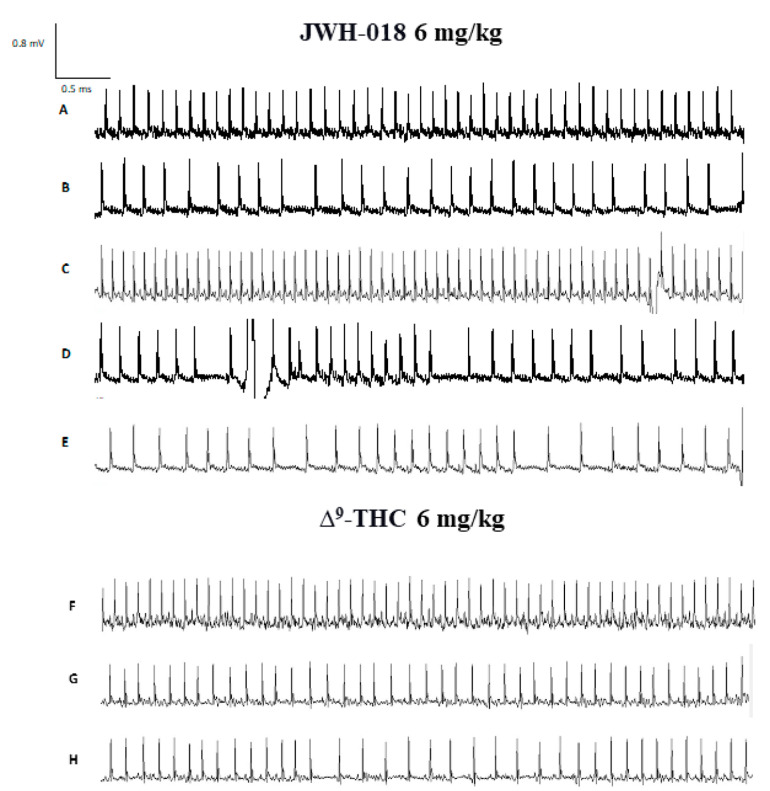
ECG track of mice treated with JWH-018 (6 mg/kg) ECG track representing basal (panel (**A**)), bradycardia (panel (**B**)), tachycardia (panel (**C**)) and arrhythmias (panels (**D**,**E**)). ECG track of mice treated with ∆^9^-THC (6 mg/kg). ECG track representing basal (panel (**F**)), bradycardia (panel (**G**)), and arrhythmias (panel (**H**)). Recording was performed with ecgTUNNEL system (Emka Technologies), and tracks was exported after analysis with *iox2* software.

**Figure 4 ijms-24-01631-f004:**
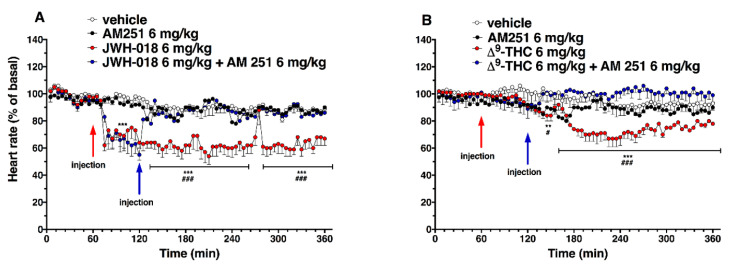
Interaction of JWH-018 (6 mg/kg i.p, panel (**A**)) and ∆^9^-THC (6 mg/kg i.p, panel (**B**)) with the selective CB_1_ receptor antagonist AM 251 (6 mg/kg, i.p.) on heart rate. Data are expressed as percentage of basal values in the form MEAN ± SEM of 8 different evaluations for each group. Statistical analysis was performed by two-way ANOVA, followed by Bonferroni’s test for multiple comparisons. ** *p* < 0.01 *** *p* < 0.001 versus vehicle, and # *p* < 0.05 ### *p* < 0.001 versus AM 251 + agonists.

**Figure 5 ijms-24-01631-f005:**
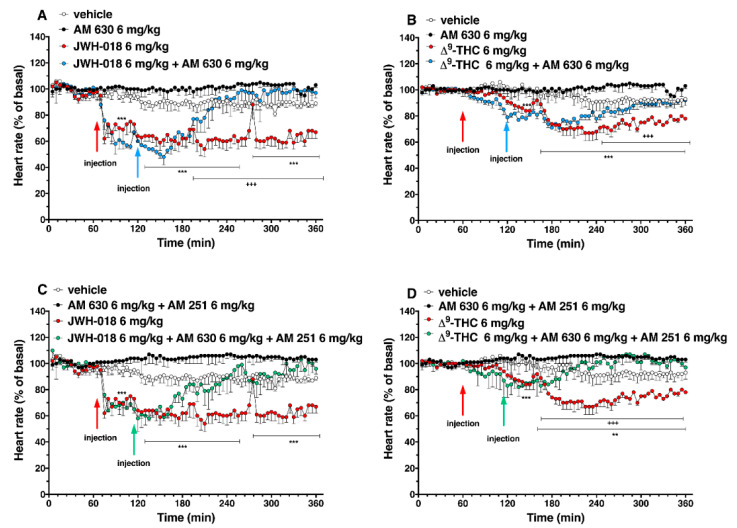
Interaction of JWH-018 (6 mg/kg i.p, panel (**A**)) and ∆^9^-THC (6 mg/kg i.p, panel (**B**)) with the selective CB_2_ receptor antagonist AM 630 (6 mg/kg, i.p.) on heart rate. Interaction of JWH-018 (6 mg/kg i.p, panel (**C**)) and ∆^9^-THC (6 mg/kg i.p, panel (**D**)) with the administration of the mixture of selective CB_1_ (AM 251; 6 mg/kg, i.p.) and CB_2_ (AM 630; 6 mg/kg, i.p.) receptor antagonists on heart rate. Data are expressed as percentage of basal values in the form MEAN ± SEM of 8 different evaluations for each group. Statistical analysis was performed by two-way ANOVA, followed by Bonferroni’s test for multiple comparisons. ** *p* < 0.01 *** *p* < 0.001 versus vehicle and +++ *p* < 0.001 versus AM 630 + agonists or AM 630 and AM 251 + agonists.

**Figure 6 ijms-24-01631-f006:**
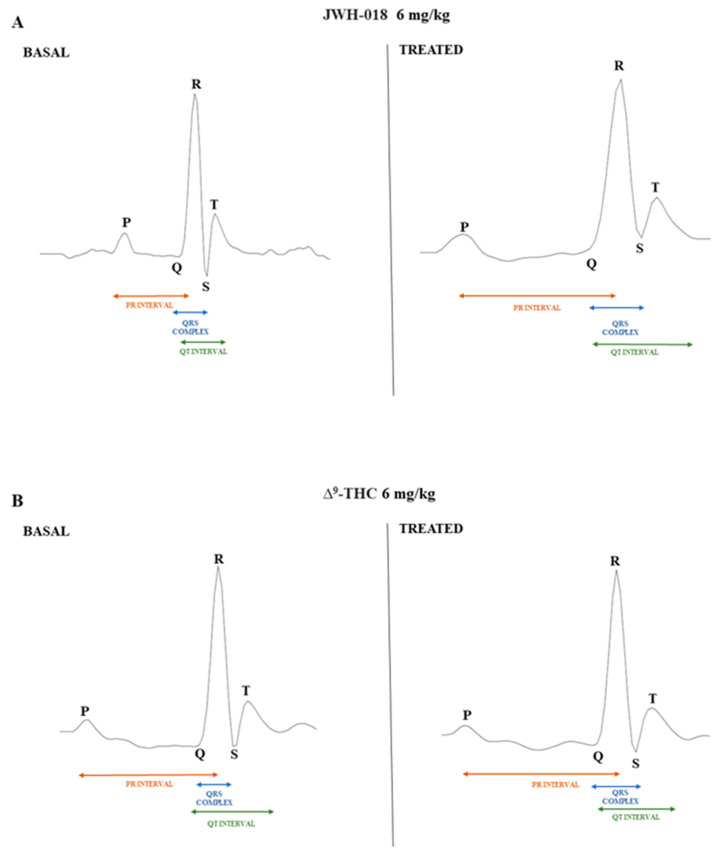
ECG waves after vehicle and JWH-018 (6 mg/kg) treatment (panel (**A**)) and ECG waves after vehicle and ∆^9^-THC (6 mg/kg) treatment (panel (**B**)). Waves were recorded with ecgTUNNEL system (Emka Technologies) and exported after analysis with *iox2* software.

**Figure 7 ijms-24-01631-f007:**
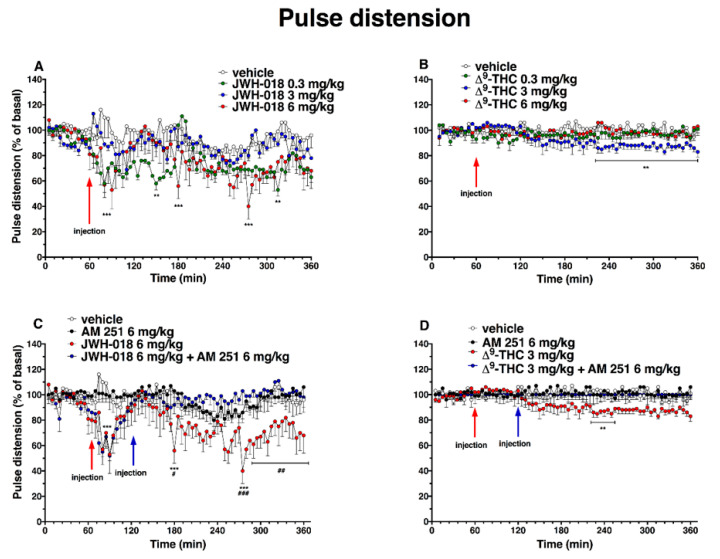
Effect of systemic administration of JWH-018 (0.3–6 mg/kg, panel (**A**)) and ∆^9^-THC (0.3–6 mg/kg, panel (**B**)) on pulse distension. Data are expressed as percentage of basal values in the form MEAN ± SEM of 6 different evaluations for JWH-018 experiments and MEAN ± SEM of 8 different evaluations for JWH-018 experiments. Statistical analysis was performed by two-way ANOVA, followed by Bonferroni’s test for multiple comparisons. ** *p* < 0.01 *** *p* < 0.001 versus vehicle. Interaction of JWH-018 (6 mg/kg i.p, panel (**C**)) and ∆^9^-THC (6 mg/kg, panel (**D**)) with the selective CB_1_ receptor antagonist AM 251 (6 mg/kg, i.p.). Data is expressed as percentage of basal values in the form MEAN ± SEM of 8 different evaluations for each group. Statistical analysis was performed by two-way ANOVA, followed by Bonferroni’s test for multiple comparisons. ** *p* < 0.01 *** *p* < 0.001 versus vehicle and # *p* <0.05, ## *p* <0.01, ### *p* < 0.001 versus AM 251 + agonists.

**Figure 8 ijms-24-01631-f008:**
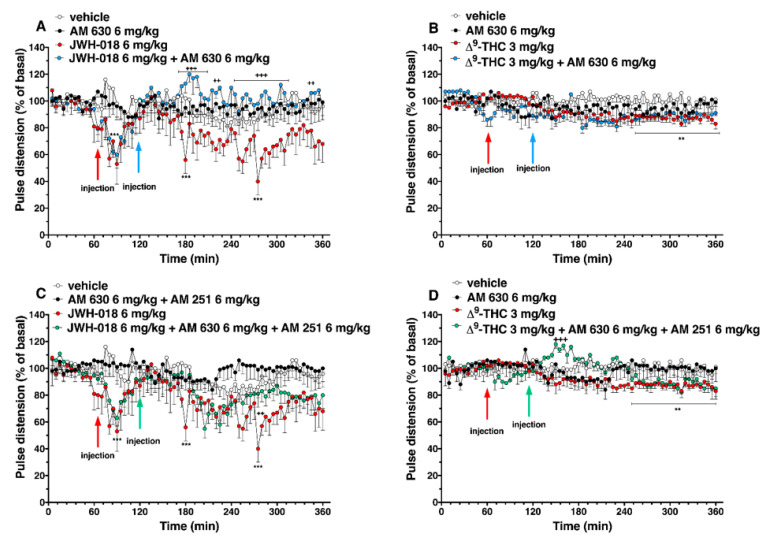
Interaction of JWH-018 (6 mg/kg i.p, panel (**A**)) and ∆^9^-THC (6 mg/kg i.p, panel (**B**)) with the selective CB_2_ receptor antagonist AM 630 (6 mg/kg, i.p.) on pulse distension. Interaction of JWH-018 (6 mg/kg i.p, panel (**C**)) and ∆^9^-THC (6 mg/kg i.p, panel (**D**)) with the administration of the mixture of selective CB_1_ (AM 251; 6 mg/kg, i.p.) and CB_2_ (AM 630; 6 mg/kg, i.p.) receptor antagonists on pulse distension. Data are expressed as percentage of basal values in the form MEAN ± SEM of 8 different evaluations for each group. Statistical analysis was performed by two-way ANOVA, followed by Bonferroni’s test for multiple comparisons ** *p* < 0.01 *** *p* < 0.001 versus vehicle and ++ *p* < 0.01, +++ *p* < 0.001 versus AM 630 + agonists or AM 630 and AM 251 + agonists.

**Figure 9 ijms-24-01631-f009:**
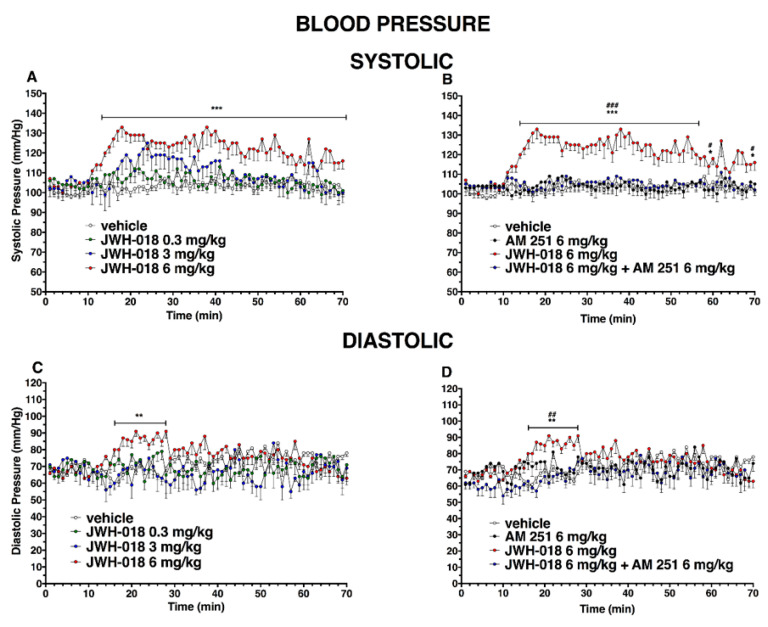
Effect of systemic administration of JWH-018 (0.3–6 mg/kg on systolic (panel (**A**)) and diastolic (panel (**C**))) blood pressure. Data are expressed as absolute values (mm/Hg) in the form MEAN ± SEM of 8 different evaluations for each group. Statistical analysis was performed by two-way ANOVA, followed by Bonferroni’s test for multiple comparisons. ** *p* < 0.01 *** *p* < 0.001 versus vehicle. Interaction of JWH-018 (6 mg/kg i.p) with the selective CB_1_ receptor antagonist AM 251 (6 mg/kg, i.p.) on systolic (panel (**B**)) and diastolic (panel (**D**)) blood pressure. Statistical analysis was performed by two-way ANOVA, followed by Bonferroni’s test for multiple comparisons. * *p* < 0.05 ** *p* < 0.01 *** *p* < 0.001 versus vehicle, # *p* < 0.05, ## *p* < 0.01, ### *p* < 0.001 versus AM 251 + agonists.

**Figure 10 ijms-24-01631-f010:**
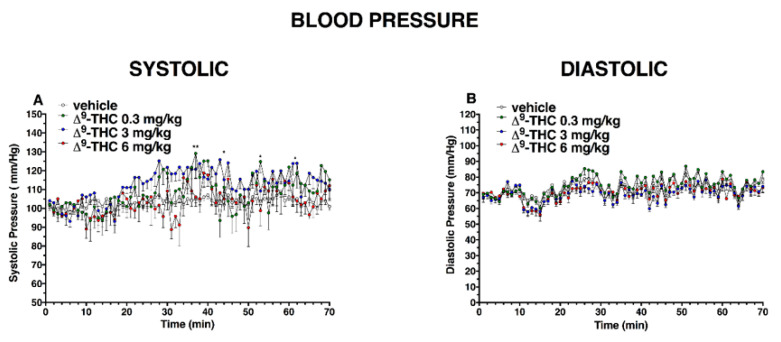
Effect of systemic administration of ∆^9^-THC (0.3–6 mg/kg on systolic (panel (**A**)) and diastolic (panel (**B**)) blood pressure. Data are expressed as absolute values (mm/Hg) in the form MEAN ± SEM of 8 different evaluations for each group. Statistical analysis was performed by two-way ANOVA, followed by Bonferroni’s test for multiple comparisons. * *p* < 0.05 ** *p* < 0.01 versus vehicle.

**Figure 11 ijms-24-01631-f011:**
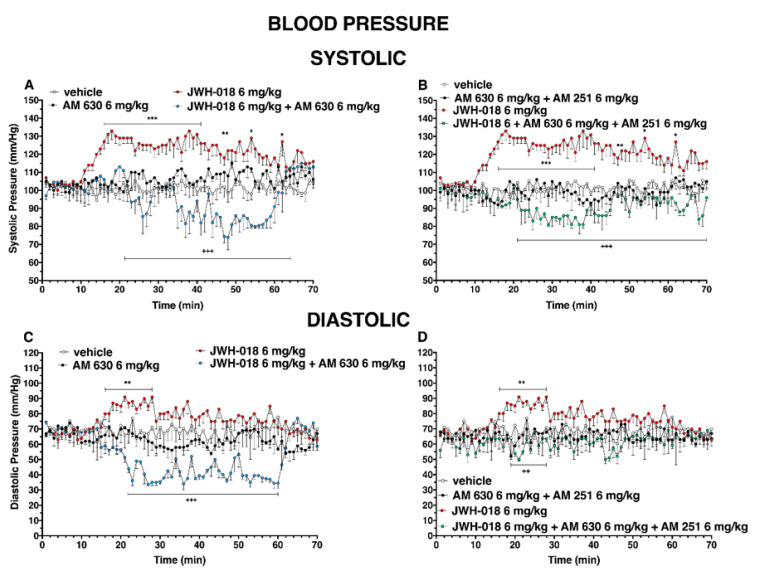
Interaction of JWH-018 (6 mg/kg i.p) with the selective CB_2_ receptor antagonist AM 630 (6 mg/kg, i.p.) and with the administration of the mixture of the selective CB_2_ receptor antagonist AM 630 (6 mg/kg) and AM 251 (6 mg/kg) on systolic (panels (**A**,**B**) respectively) and diastolic (panel (**C**,**D**), respectively) blood pressure. Statistical analysis was performed by two-way ANOVA, followed by Bonferroni’s test for multiple comparisons. * *p* < 0.05 ** *p* < 0.01 *** *p* < 0.001 versus vehicle and ++ *p* < 0.01, +++ *p* < 0.001 versus AM 630 + agonists or AM 630 and AM 251 + agonists.

**Figure 12 ijms-24-01631-f012:**
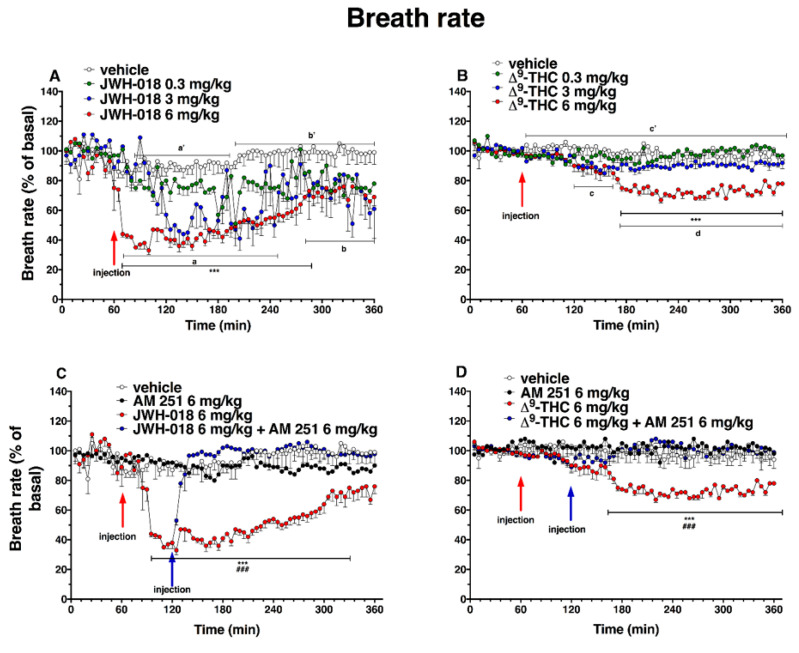
Effect of systemic administration of JWH-018 (0.3–6 mg/kg, panel (**A**)) and ∆^9^-THC (0.3–6 mg/kg, panel (**B**)) on breath rate. Data are expressed as percentage of basal values in the form MEAN ± SEM of 6 different evaluations for JWH-018 experiments and MEAN ± SEM of 8 different evaluations for ∆^9^-THC experiments. Statistical analysis was performed by two-way ANOVA, followed by Bonferroni’s test for multiple comparisons. *** *p* < 0.001 versus vehicle. Vehicle and 6 mg/kg dose curves are marked by letters (a’, b’, and c’ for vehicle and a, b, c, and d for 6 mg/kg treatment) to compare effects with plethysmography electrical parameters (Table 2). Interaction of JWH-018 (6 mg/kg i.p, panel (**C**)) and ∆^9^-THC (6 mg/kg i.p, panel (**D**)) with the selective CB_1_ receptor antagonist AM 251 (6 mg/kg, i.p.). Data are expressed as percentage of basal values in the form MEAN ± SEM of 8 different evaluations for each group. Statistical analysis was performed by two-way ANOVA, followed by Bonferroni’s test for multiple comparisons. *** *p* < 0.001 versus vehicle and ### *p* < 0.001 versus AM 251 + agonists.

**Figure 13 ijms-24-01631-f013:**
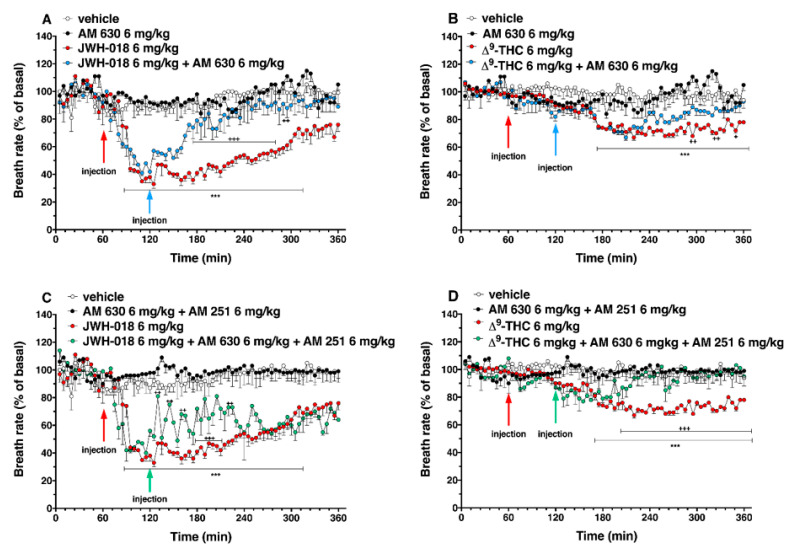
Interaction of JWH-018 (6 mg/kg i.p, panel (**A**)) and ∆^9^-THC (6 mg/kg i.p, panel (**B**)) with the selective CB_2_ receptor antagonist AM 630 (6 mg/kg, i.p.) on breath rate. Interaction of JWH-018 (6 mg/kg i.p, panel (**C**)) and ∆^9^-THC (6 mg/kg i.p, panel (**D**)) with the administration of the mixture of selective CB_1_ (AM 251; 6 mg/kg, i.p.) and CB_2_ (AM 630; 6 mg/kg, i.p.) receptor antagonists on breath rate. Data are expressed as percentage of basal values in the form MEAN ± SEM of 8 different evaluations for each group. Statistical analysis was performed by two-way ANOVA, followed by Bonferroni’s test for multiple comparisons *** *p* < 0.001 versus vehicle, and + *p* < 0.05, ++ *p* < 0.01, +++ *p* < 0.001 versus AM 630 + agonists or AM 630 and AM 251 + agonists.

**Figure 14 ijms-24-01631-f014:**
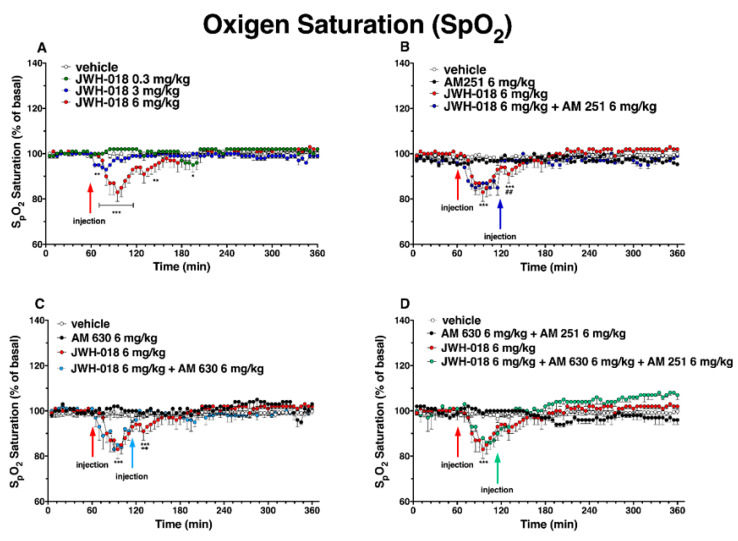
Effect of systemic administration of JWH-018 (0.3–6 mg/kg, panel (**A**)) on arterial saturation. Data are expressed as percentage of basal values in the form MEAN ± SEM of 6 different evaluations for JWH-018. Statistical analysis was performed by two-way ANOVA, followed by Bonferroni’s test for multiple comparisons. * *p* < 0.05 ** *p* < 0.01 *** *p* < 0.001 versus vehicle. Interaction of JWH-018 (6 mg/kg i.p.) with the selective CB_1_ receptor antagonist AM 251 (6 mg/kg, i.p., panel (**B**)), with the selective CB_2_ receptor antagonist AM 630 (6 mg/kg, i.p., panel (**C**)), with the administration of the mixture of AM 630 and AM 251 (panel (**D**)). Data are expressed as percentage of basal values in the form MEAN ± SEM of 8 different evaluations for each group. Statistical analysis was performed by two-way ANOVA, followed by Bonferroni’s test for multiple comparisons. * *p* < 0.05 ** *p* < 0.01 *** *p* < 0.001 versus vehicle, and ## *p* < 0.01 versus AM 251 + agonists, and ++ *p* < 0.01 versus AM 630 + agonists or AM 630 and AM 251 + agonists.

**Table 1 ijms-24-01631-t001:** Effect of systemic administration of vehicle (a’, b’), JWH-018 6 mg/kg (a and b) and ∆^9^-THC 6 mg/kg on cardiac electrical parameters (ECG analysis). Data are relative to HR dose–response curve (Figure 2A,B). Data are expressed as mean overall effects of 15 evaluations for baseline and after vehicle or JWH-018 treatment of 4 mice. Data were analyzed with *iox2* software, and they are expressed as MEAN ± SEM of evaluations recorded in 1-min time window, capturing data with an acquisition frequency of 1000 Hz. Statistical analysis was performed by Student’s *t*-test for each basal and after-treating parameter comparisons. * *p* < 0.05 ** *p* < 0.01 *** *p* < 0.001 vs. basal values. HR (bpm; Heart Rate), RR (ms; time between two consecutive peaks), PR (ms; time from the onset of the *p* wave to the start of the QRS complex), QRS (ms; is the time from the start to the end of the QRS complex), QT (ms; time from the beginning of the QRS complex to the end of the T wave), QTcF (QT/∛ RR).

ECG Parameters													
Vehicle of JWH-018							Vehicle of ∆^9^-THC						
	HR	RR	PR	QRS	QT	QTcF		HR	RR	PR	QRS	QT	QTcF
Basal	681.61 ± 4.40	88.43 ± 0.58	21.07 ± 2.98	9.78 ± 0.32	25.27 ± 0.97	61.38 ±3.45	Basal	708.92 ± 1.51	85.77 ± 0.94	21.26 ± 2.47	9.32 ± 0.31	24.57 ± 0.78	55.94 ± 1.81
a’	680.47 ± 3.55	88.55 ± 0.46	21.87 ± 2.96	9.03 + 0.30	25.62 ± 0.79	60.62 ± 2.94	a’	709.92 ± 1.97	85.60 ± 0.91	21.40 ± 2.51	9.50 ± 0.33	25.05 ± 0.34	57.18 ± 1.94
b’	615.63 ± 4.05 ***	97.89 ± 0.62 ***	21.48 ± 2.62	9.09 ± 0.36	25.14 ± 0.79	61.57 ± 2.48	b’	645.22 ± 2.98 ***	93.56 ± 0.92 ***	21.50 ± 3.14	9.41 ± 0.36	24.44 ± 1.01	56.67 ± 2.45
JWH-018 6 mg/kg							∆^9^-THC 6 mg/kg						
Basal	653.36 ± 4.50	92.14 ± 0.61	18.06 ± 1.16	9.71 ± 0.09	24.73 ± 0.23	54.86 ± 0.57	Basal	689.52 ± 7.04	87.84 ± 0.95	21.66 ± 0.65	9.34 ± 0.24	25.07 ± 0.42	56.72 ± 2.57
a	346.43 ± 12.21 ***	181.50 ± 4.77 ***	18.84 ± 0.97	10.12 ± 0.17 ***	27.16 ± 0.42 ***	48.38 ± 1.36 ***	a	604.98 ± 7.48 ***	96.30 ± 1.67 ***	21.98 ± 0.54	9.56 ± 0.21 *	25.32 ± 0.41	55.13 ± 2.49
b	759.82 ± 4.10 ***	79.02 ± 4.50 ***	24.37 ± 0.89 ***	9.79 ± 0.16	25.33 ± 0.39 ***	58.63 ± 1.30 ***	b	540.87 ± 6.33 ***	112.28 ± 0.95 ***	22.09 ± 0.40 *	9.62 ± 0.23 **	25.38 ± 0.45	54.81 ± 2.31

**Table 2 ijms-24-01631-t002:** Effect of systemic administration of vehicle (a’, b’), JWH-018 6 mg/kg (a and b) and ∆^9^-THC 6 mg/kg on respiratory electrical parameters (plethysmography analysis). Data are relative to BR dose-response curve (Figure 12A,B). Data are expressed as mean overall effects of 15 evaluations for baseline and after vehicle or JWH-018 treatment of 4 mice. Data are analyzed with *iox2* software, and they are expressed as MEAN ± SEM of evaluations recorded in 1-min time window, capturing data with an acquisition frequency of 1000 Hz. Statistical analysis was performed by Student’s *t*-test for each basal and after treating parameter comparisons. ** *p* < 0.01, *** *p* < 0.001 vs. basal values. FR_IE (bpm; breathing frequency); BB_IE (ms; breath length computed by adding inspiration and expiration duration); TV (mL/s; tidal inspiration volume); EXP/INSP_V_RATIO__AVER (ratio of expired volume/tidal volume); RT (ms; Relaxion Time)).

Plethysmography Parameters											
Vehicle of JWH-018						Vehicle of ∆^9^-THC					
	fr_ie	BB_ie	TV	exp/insp_V ratio__aver	RT		fr_ie	BB_ie	TV	exp/insp_V ratio__aver	RT
Basal	230.83 ± 1.99	261.01 ± 2.77	0.15 ± 0.01	0.07 ± 1.15	116.79 ± 3.10	Basal	270.59 ± 3.41	223.62 ± 2.97	0.17 ± 0.01	−0.05 ± 0.78	83.73 ± 1.40
a’	201.47 ± 2.54 ***	299.94 ± 4.01 ***	0.15 ± 0.01	0.09 ± 0.79	117.90 ± 1.93	a’	268.57 ± 3.70	225.36 ± 2.17	0.18 ± 0.01	−0.08 ± 0.97	84.55 ± 1.79
b’	230.48 ± 5.56	263.55 ± 5.51	0.16 ± 0.01	−0.04 ± 0.97	114.70 ± 2.94						
JWH-018 6 mg/kg						∆^9^-THC 6 mg/kg					
Basal	249.51 ± 5.51	243.64 ± 4.56	0.13 ± 0.003	0.06 ± 1.11	107.19 ± 4.36	Basal	256.09 ± 2.63	239.25 ± 3.83	0.18 ± 0.01	−0.038 ± 0.63	93.65 ± 3.26
a	138.98 ± 2.07 ***	433.77 ± 8.03 ***	0.11 ± 0.002 ***	−0.07 ± 0.97	170.28 ± 9.02 ***	a	219.31 ± 2.39 ***	276.23 ± 2.78 ***	0.18 ± 0.002	0.02 ± 0.69	114.08 ± 2.44 ***
b	170.96 ± 2.76 ***	353.61 ± 6.39 **	0.12 ± 0.002 ***	−0.07 ± 2.42	161.15 ± 6.35 ***	b	178.49 ± 2.07 ***	341.73 ± 4.37 ***	0.17 ± 0.002 **	0.07 ± 0.92	131.05 ± 4.69 ***

## Data Availability

The data presented in this study are available on request from the first (Beatrice Marchetti) and corresponding author (Matteo Marti) for researchers of academic institutes who meet the criteria for access to the confidential data.

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
