# Peer review of "The Old and the New: Cardiovascular and Respiratory Alterations Induced by Acute JWH-018 Administration Compared to Δ9-THC—A Preclinical Study in Mice"

_ijms, 2023, doi:10.3390/ijms24021631_

Round 1

Reviewer 1 Report

The current study investigates cardiac (heart rate) alterations and electrocardiogram (ECG) parameters (PR interval, QRS complex duration, QT and QTc interval), together with respiratory (oxygen saturation, breath rate and plethysmography parameters such as breathing frequency, breath length, tidal volume, ratio of expired volume/tidal volume and relaxation time), and vascular (pulse distension, systolic and diastolic pressure) alterations induced by an acute administration of a synthetic cannabinoid JWH-018 (0.3-3-6 mg/kg) in awake CD-1 male mice. To better evaluate the potency of JWH-018 effects, Δ9-THC was used (0.3-3-6 mg/kg) for comparison. Moreover, to confirm the involvement of CB1 receptor activation in cardiovascular and respiratory changes the CB1 receptor antagonist AM 251 was tested.

Major comments

For the clarity of reading Authors please move sections related vascular alterations after cardiac alterations and ECG parameters.

Lines 722-724. "Specifically, all JWH-018- and Δ9-THC-induced  effects can be primarily attributed to CB1 receptors binding since these outcomes were  totally prevented by pretreatment with AM 251, a selective CB1 receptors antagonist". There is a discrepancy between this statement and information provided in the section Methods as well as data shown on figures. Lines 629-631: AM 251 (6 mg/kg) was administered 60 minutes after JWH-018 and Δ9-THC injection in cardiovascular experiments and 20 minutes before JWH-018 and Δ9-THC injections in blood pressure analyses. From Fig. 9 B and D it appears that AM 251 was administered at the same time as agonists. Cardiovascular experiments – I do not understand a rationale underlaying administration of AM 251 one hour after injection of agonists.

Minor comments

Line 149. Should be: at a lowest dose.

Fig.  4B. Should be: Δ9-THC 6 mg/kg + AM 251 6 mg/kg.

Fig. 4B,  Fig. 6B, Fig. 7B, Fig. 8C I strongly suggest to replace a symbol (open circle) °p<0.05 °°°p < 0.001 versus.., as this symbol is used in a graph of vehicle groups.

Table 1, Table 3 *** p<0.001 versus ??

Table 2. * p<0.05, ** p<0.01, *** p<0.001 versus ??

Table 4. ** p<0.01 versus ??

Fig. 7B. Should be JWH-019 6 mg/kg; JWH-018 6 mg/kg+ AM 251 6 mg/kg.

Fig. 9. Replace mg/Kg with mg/kg. * p<0.05 not shown on the figure. The legend lacks °° p<0.01. Should be agonist.

Line 597. .. (light period from 6:30 AM to 6:30 PM); line 601… (dark between 7 p.m.–7 a.m.)??

Author Response

Response to Reviewer 1

We thank the Reviewer 1 for his/her evaluation of our manuscript and for helpful concerns to improve the article. In this revised version of the work we have addressed the major concerns of the referee (the parts are highlighted in yellow).

The current study investigates cardiac (heart rate) alterations and electrocardiogram (ECG) parameters (PR interval, QRS complex duration, QT and QTc interval), together with respiratory (oxygen saturation, breath rate and plethysmography parameters such as breathing frequency, breath length, tidal volume, ratio of expired volume/tidal volume and relaxation time), and vascular (pulse distension, systolic and diastolic pressure) alterations induced by an acute administration of a synthetic cannabinoid JWH-018 (0.3-3-6 mg/kg) in awake CD-1 male mice. To better evaluate the potency of JWH-018 effects, Δ9-THC was used (0.3-3-6 mg/kg) for comparison. Moreover, to confirm the involvement of CB1 receptor activation in cardiovascular and respiratory changes the CB1 receptor antagonist AM 251 was tested.

Major comments

Rev1Q1: For the clarity of reading Authors please move sections related vascular alterations after cardiac alterations and ECG parameters.

AA: We thank the Reviewer 1 for pointing out this inconsistency and we moved sections to change the order.

Rev1Q2: Lines 722-724. "Specifically, all JWH-018- and Δ9-THC-induced effects can be primarily attributed to CB1 receptors binding since these outcomes were totally prevented by pretreatment with AM 251, a selective CB1 receptors antagonist". There is a discrepancy between this statement and information provided in the section Methods as well as data shown on figures. Lines 629-631: AM 251 (6 mg/kg) was administered 60 minutes after JWH-018 and Δ9-THC injection in cardiovascular experiments and 20 minutes before JWH-018 and Δ9-THC injections in blood pressure analyses. From Fig. 9 B and D it appears that AM 251 was administered at the same time as agonists. Cardiovascular experiments – I do not understand a rationale underlaying administration of AM 251 one hour after injection of agonists.

AA: We thank the Reviewer 1 for highlighting this important point concerning the experimental protocol carried out in this session. First of all, we modified the current line 860 (722 before modifications) to avoid inconsistency. Moreover, we would like to clarify the rational underlaying the different administration time points of AM-251. In particular, Mouse Ox Plus system provide data from a long lasting (5 hours) registration session during which the mouse can freely move in the cage. Thus, subsequent administration of the compounds and observation of the effect of the co-administration were possible in an adequate time interval. Moreover, the choice to administrate AM-251 later than the agonist is based on the attempt of simulating clinical intoxications management. On the other hand, experimental protocol carried out through BP2000 system provides a registration for 70 minutes only during which the animal is restrained. Therefore, the antagonist has been injected 20 minutes before the administration of the agonist in order to decrease the stress levels (reducing manipulation) and providing the longest possible analysis of the co-administration.

Minor comments

Rev1Q3: Line 149. Should be: at a lowest dose.

AA: We thank the Reviewer 1 for pointing out this inaccuracy and modified line 153.

Rev1Q4: Fig. 4B. Should be: Δ9-THC 6 mg/kg + AM 251 6 mg/kg.

AA: We thank the Reviewer 1 for highlighting this inaccuracy and modified Fig 4B.

Rev1Q5: Fig. 4B, Fig. 6B, Fig. 7B, Fig. 8C I strongly suggest to replace a symbol (open circle) °p<0.05 °°°p < 0.001 versus.., as this symbol is used in a graph of vehicle groups.

AA: We thank Reviewer 1 for this suggestion. We convert ° (open circle) symbol to # symbol in all graphs.

Rev1Q6: Table 1, Table 3 *** p<0.001 versus ??

AA: We thank the Reviewer 1 for pointing out this inaccuracy and we provided to modify Table1 and 3 captions.

Rev1Q7: Table 2. * p<0.05, ** p<0.01, *** p<0.001 versus ??

AA: We thank the Reviewer 1 for pointing out this inaccuracy and we provided to modify Table 2 caption.

Rev1Q8: Table 4. ** p<0.01 versus ??

AA: We thank the Reviewer 1 for pointing out this inaccuracy and we provided to modify Table 4 caption.

Rev1Q9: Fig. 7B. Should be JWH-019 6 mg/kg; JWH-018 6 mg/kg+ AM 251 6 mg/kg.

AA: We thank the Reviewer 1 and modified figure 7B.

Rev1Q10: Fig. 9. Replace mg/Kg with mg/kg. * p<0.05 not shown on the figure. The legend lacks °° p<0.01. Should be agonist.

AA: We thank the Reviewer 1 for pointing out this inaccuracy. We converted mg/Kg in mg/kg, we checked figure caption and corrected statistical symbol legend.

Rev1Q11: Line 597. .. (light period from 6:30 AM to 6:30 PM); line 601… (dark between 7 p.m.–7 a.m.)??

AA: We thank the Reviewer 1 for highlighting this repetition. We provided to eliminate unnecessary sentence.

Reviewer 2 Report

This study investigates alterations in heart rate and electrocardiogram parameters, as well as some respiratory parameters and vascular alterations induced by acute administration of JWH-018 in male mice and compares the results with those produced by Δ9-THC.

In addition, AM 251, a CB1 receptor antagonist, was used to confirm that the onset of the alterations is due to the activation of the CB1 receptor.

I can't comment too much on the physiological part of the paper as I'm not an expert in these techniques, although everything seems correct to me. My expertise is in cannabinoid receptors and their interactions with other receptors.

The cannabinoid CB1 and CB2 receptors are present in cardiovascular tissues, and JWH-018 is a full agonist of both cannabinoid receptors.

The results seem to indicate that the alterations are mediated by the CB1 receptor on cardiac β-adrenoceptors, but the effects of activating the CB2 receptor cannot be ruled out.

This paper should be completed with the administration of AM 630, a CB2 receptor antagonist, and with a mixture of the antagonists AM 251 and AM 630.

Obviously, the CB2 receptor must be considered in the introduction and the corresponding results in the discussion.

Until such experiments have been carried out, the paper should not be published.

Author Response

Response to Reviewer 2

We thank the Reviewer 2 for his/her evaluation of our manuscript and for helpful concerns to improve the article. In this revised version of the work we have addressed the major concerns of the referee (the parts are highlighted in green).

Rev2Q1: This study investigates alterations in heart rate and electrocardiogram parameters, as well as some respiratory parameters and vascular alterations induced by acute administration of JWH-018 in male mice and compares the results with those produced by Δ9-THC.

In addition, AM 251, a CB1 receptor antagonist, was used to confirm that the onset of the alterations is due to the activation of the CB1 receptor.

I can't comment too much on the physiological part of the paper as I'm not an expert in these techniques, although everything seems correct to me. My expertise is in cannabinoid receptors and their interactions with other receptors.

The cannabinoid CB1 and CB2 receptors are present in cardiovascular tissues, and JWH-018 is a full agonist of both cannabinoid receptors.

The results seem to indicate that the alterations are mediated by the CB1 receptor on cardiac β-adrenoceptors, but the effects of activating the CB2 receptor cannot be ruled out.

This paper should be completed with the administration of AM 630, a CB2 receptor antagonist, and with a mixture of the antagonists AM 251 and AM 630.

Obviously, the CB2 receptor must be considered in the introduction and the corresponding results in the discussion.

Until such experiments have been carried out, the paper should not be published.

AA: We thank the reviewer 2 for the suggestion, and we provide further data to complete with CB2 receptor antagonist. We also complete the full manuscript, expanding the sections of abstract, introduction, discussion and method and materials, in addition of that of results.

Reviewer 3 Report

In the present manuscript Marchetti et al reported an interesting study comparing the cardio-respiratory effects of the synthetic cannabinoid JWH-018 and THC, the psychoactive compound of marijuana. Both cannabinoid act on cannabinoid receptor type-1 (CB1) provoking marked effects as shown by alterations in heart rates, conduction and breathing rate as well as blood pressure and oxygen saturation. These drugs or their vehicle were administered parenterally to conscious mice and cardio-respiratory function was then recorded up to 6hrs.  The authors have revealed significant and detrimental changes induces by JWH-018 administration, especially at higher doses when compared to THC. Unfavorable effects of CB1 activation were abolished by AM-251 pretreatment. The findings are clear, the paper is technically sound. As for the formal aspects, this the manuscript is well-written and readable, however, there are some typos that should be corrected in the final version.

Nevertheless, there are some major comments need to be addressed:

1.    Based on the pharmacokinetics of JHW-018, it is a full agonist for both CB1 and CB2 receptors (PMID:10940540 and 25921407). Why did the authors choose this synthetic cannabinoid instead of using novel, more potent and specific CB1 agonists? 

2.    Did the authors see any CB2-related effect on cardiac function if any expected?

3.    Based on Fig2 and Fig9-10, the vascular effects of CB1 activation is proximal compared to the cardiac effects as seen by an early increase of systolic/diastolic blood pressure induced by JWH-018 treatment. However, it should be independent of the activation of sympathetic tone, since the heart rate remained unchanged. What could be the possible explanation for this phenomenon?

4.    What is the primary source of CB1 expression in the cardiovascular system? 

5.    There’s a considerable reduction of pulse distension in these mice induced by both JWH-018 and THC treatment that was reversed by AM-251 administration. Did the authors have any data regarding the cardiac function as well? How cardiac output would be affected by CB1 receptor activation?

6.    THC was able to reduce breathing rate (please see Fig.6A). How did THC treatment affect blood oxygen saturation?

Minor comments:

1.    Please consider to reduce the resolution of your data by using less points (e.g.: showing data points at 0-60-120-180 min etc. only). That would help to get a clearer picture about the effect of the drugs.

2.    Please consider to merge Table1 and 2 as well as Table3 and 4 thereby the readers can easily compare the effects of synthetic cannabinoids and THC.   

Author Response

Response to Reviewer 3

We thank the Reviewer 3 for his/her evaluation of our manuscript and for helpful concerns to improve the article. In this revised version of the work we have addressed the major concerns of the referee (the part are highlighted in blue).

In the present manuscript Marchetti et al reported an interesting study comparing the cardio-respiratory effects of the synthetic cannabinoid JWH-018 and THC, the psychoactive compound of marijuana. Both cannabinoid act on cannabinoid receptor type-1 (CB1) provoking marked effects as shown by alterations in heart rates, conduction and breathing rate as well as blood pressure and oxygen saturation. These drugs or their vehicle were administered parenterally to conscious mice and cardio-respiratory function was then recorded up to 6hrs. The authors have revealed significant and detrimental changes induces by JWH-018 administration, especially at higher doses when compared to THC. Unfavorable effects of CB1 activation were abolished by AM-251 pretreatment. The findings are clear, the paper is technically sound. As for the formal aspects, this the manuscript is well-written and readable, however, there are some typos that should be corrected in the final version.

Nevertheless, there are some major comments need to be addressed:

Rev3Q1: 1. Based on the pharmacokinetics of JHW-018, it is a full agonist for both CB1 and CB2 receptors (PMID:10940540 and 25921407). Why did the authors choose this synthetic cannabinoid instead of using novel, more potent and specific CB1 agonists?

AA: We thank the reviewer 3 for this interesting question. The NPS illegal market constantly change, introducing newer derivatives of already known substances (UNODC, 2013; Luethi and Liechti, 2020). In particular, Synthetic Cannabinoids have evolved to the fourth generation (Malaca et al., 2022). Nevertheless, JWH-018, a first-generation Synthetic Cannabinoid, was the founder molecule of this class of NPS and most of subsequent Synthetic Cannabinoids are structurally and functionally related to this reference drug (Alves et al., 2020). Nowadays, JWH-018 is still widely used and identified by laboratory screening in many cases of intoxication (Darke et al., 2019; Theunissen et al., 2021). While a specific preclinical study on the cardio-respiratory effects of THC and SCs showing the different parameters is not yet provided. Therefore, this novel pioneering study for the characterization of cardiovascular and respiratory effects induced by JWH-018 in comparison to THC could be the basis for more current SCBs screening.

  1. United Nations Office on Drugs and Crime (UNODC). The Challenge of New Psychoactive Substances: A Report from the Global SMART Programme: Vienna, 2013. Available online: https://www.unodc.org/documents/scientific/NPS_2013_SMART.pdf

  2. Luethi, D.; Liechti, M.E. Designer drugs: mechanism of action and adverse effects. Arch Toxicol. 2020, 94(4), 1085-1133. doi:10.1007/s00204-020-02693-7

  3. Malaca, S., Busardò, F.P., Nittari, G., Sirignano, A., Ricci, G. Fourth Generation of Synthetic Cannabinoid Receptor Agonists: A Review on the Latest Insights. Curr Pharm Des. 2022, 28(32),2603-2617. doi:10.2174/1381612827666211115170521. PMID: 34781870.

  4. Alves, V.L.; Gonçalves, J.L.; Aguiar, J.; Teixeira, H.M.; Câmara, J.S. The synthetic cannabinoids phenomenon: from structure to toxicological properties. A review. Crit Rev Toxicol. 2020, 50(5), 359-382. doi:10.1080/10408444.2020.1762539

  5. Theunissen, E. L., Reckweg, J. T., Hutten, N. R. P. W., Kuypers, K. P. C., Toennes, S. W., Neukamm, M. A., Halter, S., & Ramaekers, J. G. (2021). Intoxication by a synthetic cannabinoid (JWH-018) causes cognitive and psychomotor impairment in recreational cannabis users. Pharmacology, biochemistry, and behavior202, 173118. https://doi.org/10.1016/j.pbb.2021.173118

  6. Darke, S., Duflou, J., Farrell, M., Peacock, A., & Lappin, J. (2020). Characteristics and circumstances of synthetic cannabinoid-related death. Clinical toxicology (Philadelphia, Pa.), 58(5), 368–374. https://doi.org/10.1080/15563650.2019.1647344

Rev3Q2: 2. Did the authors see any CB2-related effect on cardiac function if any expected?

AA: We thank the reviewer 3 for this question, which was also suggested by another reviewer of the manuscript. Therefore, we have implemented data from the manuscript to understand a possible CB2-mediated mechanism on cardiovascular and respiratory functions. Specifically, we evaluated the effects of JWH-018 and Δ9-THC following the treatment with AM 630, a CB2 receptors selective antagonist, and the treatment with the mixture of AM 251 and AM 630.

Rev3Q3: 3. Based on Fig2 and Fig9-10, the vascular effects of CB1 activation is proximal compared to the cardiac effects as seen by an early increase of systolic/diastolic blood pressure induced by JWH-018 treatment. However, it should be independent of the activation of sympathetic tone, since the heart rate remained unchanged. What could be the possible explanation for this phenomenon?

AA: We thank the Reviewer 3 for this suggesting consideration. The Figure 2 (panel A) represent heart rate (HR) changes and the experiment was carried out recording one hour of basal (untreated mice), and subsequently JWH-018 was injected. Immediately after injection, the HR suddenly decreased to ~40%. This is probably a result of centrally and peripherally increase of vagal tone (Niederhoffer ans Szabo, 1999). Despite this, also sympathetic activity was probably involved after CB1 activation, explaining the tachycardia episodes, tachyarrhythmias and blood pressure increase (Schindler et al., 2017; Richards, 2020). Therefore, both of mechanism could be involved.

  1. Niederhoffer, N.; Szabo, B. Effect of the cannabinoid receptor agonist WIN55212-2 on sympathetic cardiovascular regulation. Br J Pharmacol. 1999, 126(2), 457-66. doi: 10.1038/sj.bjp.0702337.

  2. Schindler, C.W.; Gramling, B.R.; Justinova, Z.; Thorndike, E.B.; Baumann, M.H. Synthetic cannabinoids found in "spice" products alter body temperature and cardiovascular parameters in conscious male rats. Drug Alcohol Depend. 2017, 179, 387-394. doi: 10.1016/j.drugalcdep.2017.07.029.

  3. Richards, J.R. Mechanisms for the Risk of Acute Coronary Syndrome and Arrhythmia Associated With Phytogenic and Synthetic Cannabinoid Use. J Cardiovasc Pharmacol Ther. 2020, 25(6), 508-522. doi: 10.1177/1074248420935743.

Rev3Q4: 4. What is the primary source of CB1 expression in the cardiovascular system?

AA: As reported in introduction section “CB receptors in the cardiovascular system are localized in coronary arteries, myocardium, endothelial and smooth muscle cells. In particular, CB1 receptors are also expressed on pre-synaptic sympathetic nerve terminals innervating the cardiovascular systems [34].” The endocannabinoid system plays an important role in cardiovascular function (as contractility or vascular tone; O’Sullivan et al., 2015). In particular, in human, the endocannabinoid anandamide (AEA) decreases contractile performance of cardiac muscle (Bonz et al., 2003) via CB1 receptor stimulation. This evidence was also confirmed through preclinical studies, which demonstrated that AEA is able to diminish the contractility, inhibiting the function sodium and calcium channels (Al Kury et al., 2014). Concerning vascular functions, AEA was able to induce vasorelaxation on human isolated mesenteric arteries (Stanley and O’Sullivan, 2012) and the Ellis et al. preclinical study demonstrated that the vasorelaxation was associated to an increase in vasoactive prostanoid (Ellis et al., 1995). Despite this, our data and preclinical studies on Synthetic cannabinoids and THC showed other evidence (i.e. increase of blood pressure or tachyarrhythmias), that involve different mechanisms as increase of sympathetic outflow (Schindler et al., 2017; Dean et al., 2016; Rudź et al., 2012).

  1. O'Sullivan, S.E. Endocannabinoids and the Cardiovascular System in Health and Disease. Handb Exp Pharmacol. 2015, 231, 393-422. doi: 10.1007/978-3-319-20825-1_14.

  2. Bonz, A., Laser, M., Küllmer, S., Kniesch, S., Babin-Ebell, J., Popp, V., Ertl, G., Wagner, J.A. Cannabinoids acting on CB1 receptors decrease contractile performance in human atrial muscle. J Cardiovasc Pharmacol. 2003, 41(4), 657-64. doi: 10.1097/00005344-200304000-00020. PMID: 12658069.

  3. Stanley, C.P., Hind, W.H., Tufarelli, C., O'Sullivan, S.E. The endocannabinoid anandamide causes endothelium-dependent vasorelaxation in human mesenteric arteries. Pharmacol Res. 2016, 113(Pt A), 356-363. doi: 10.1016/j.phrs.2016.08.028. Epub 2016 Sep 12. PMID: 27633407; PMCID: PMC5113919.

  4. Ellis, E.F., Moore, S.F., Willoughby, K.A. Anandamide and delta 9-THC dilation of cerebral arterioles is blocked by indomethacin. Am J Physiol Heart Circ Physiol, 1995, 269, H1859–H1864.

  5. Schindler, C.W.; Gramling, B.R.; Justinova, Z.; Thorndike, E.B.; Baumann, M.H. Synthetic cannabinoids found in "spice" products alter body temperature and cardiovascular parameters in conscious male rats. Drug Alcohol Depend. 2017, 179, 387-394. doi: 10.1016/j.drugalcdep.2017.07.029.

  6. Dean, C.; Hillard, C.J.; Seagard, J.L.; Hopp, F.A.; Hogan, Q.H. Components of the cannabinoid system in the dorsal periaqueductal gray are related to resting heart rate. Am J Physiol Regul Integr Comp Physiol. 2016, 311(2), R254-62. doi: 10.1152/ajpregu.00154.2016.

  7. Rudź, R.; Schlicker, E.; Baranowska, U.; Marciniak, J.; Karabowicz, P.; Malinowska, B. Acute myocardial infarction inhibits the neurogenic tachycardic and vasopressor response in rats via presynaptic cannabinoid type 1 receptor. J Pharmacol Exp Ther. 2012, 343(1), 198-205. doi: 10.1124/jpet.112.196816.

Rev3Q5: 5. There’s a considerable reduction of pulse distension in these mice induced by both JWH-018 and THC treatment that was reversed by AM-251 administration. Did the authors have any data regarding the cardiac function as well? How cardiac output would be affected by CB1 receptor activation?

AA. The Mouse Ox Plus system can evaluate cardiovascular and respiratory parameters (heart rate, pulse distension, breath rate and oxygen saturation). This system could not provide evaluation of other parameters such as cardiac output or stroke volume. Moreover, the pulse distention is measured through Pulse Oximetry which measures the oxygen content of arterial blood. Blood is identified as being arterial because of its pulsatile nature. This pulsation is identifiable because it causes a cyclic change in the absorption of light energy from the red and infrared LEDs (Light Emitting Diodes) as it passes through the tissue due to the presence of changing quantities of blood that occur with every heartbeat. All other non-blood material such as skin, fat and muscle do not change light absorption level with heart rate. Additionally, non-arterial blood (venous or capillary) does not change light absorption level with heart rate either, because pulsation is essentially completely diminished by the termination of the arterioles. Thus, the pulse oximetry measurement is being made across only pulsating blood, which can only be arterial. Because this blood is arterial, it possesses systemic arterial oxygen content, which is what is measured (MouseOx Plus; 2015). Despite this, understanding how heart frequency, pulse distension and cardiac output are linked is worth noting. Cardiac output depends on factors such as heart frequency, contractility, preload whose decrease results in decrease of cardiac output. Moreover, cardiac output depends on the afterload which is closely related to arterial blood pressure and vascular tone (Vincent, 2008). An increased afterload is an index of high arterial pressure (Feher, 2012) and an increased afterload can lead to a reduction of cardiac output (Vincent, 2008). Therefore, this evidence, together with our data that indicated HR decrease and blood pressure increase, suggesting that the cardiac output in mice could decrease.

  1. Mouse Ox Plus- Small Animal Vital Sings Monitor, User Manual, STARR Life Sciences® Corp., 2015.

  2. Vincent JL. Understanding cardiac output. Crit Care. 2008;12(4):174. doi: 10.1186/cc6975. Epub 2008 Aug 22. PMID: 18771592; PMCID: PMC2575587.

  3. Feher J.,5.8 - The Cardiac Function Curve, Editor(s): Joseph Feher. Quantitative Human Physiology (Second Edition). Academic Press. 2012. Pages 556-564. ISBN 9780128008836, https://doi.org/10.1016/B978-0-12-800883-6.00052-5.

Rev3Q6: 6. THC was able to reduce breathing rate (please see Fig.6A). How did THC treatment affect blood oxygen saturation?

AA: As reported in line 518-519, ∆9-THC did not affect the arterial oxygen saturation. Thus, we decide to not report the data.

Minor comments:

Rev3Q7: 1. Please consider to reduce the resolution of your data by using less points (e.g.: showing data points at 0-60-120-180 min etc. only). That would help to get a clearer picture about the effect of the drugs.

AA: We thank the Reviewer 3 for this suggestion and we are aware of the large number of points in graphs. Thus, we provide to explain to him/her about the previous choice of inserting of all time point. Mouse Ox Plus system provide to analyze cardiac and respiratory in vivo data, such as oxygen saturation and pulse distension during 5 hours, with a measurement minute by minute. Moreover, data in the graphs consist in mean values of a wider range of measurement recorded in a minute. Therefore, all time points were inserted to provide an analysis that can show how these parameters are modified during a long-lasting period, highlighting their changes in detail. Taken together, these statements support the assumption that reducing data points could lead to misrepresented cardiovascular effects that are monitored for five hours of measurements of the two compounds.

Rev3Q8: 2. Please consider to merge Table1 and 2 as well as Table3 and 4 thereby the readers can easily compare the effects of synthetic cannabinoids and THC.

AA: We thank the Reviewer 3 for this suggestion and we merged tables. In particular Table 1 and Table 2 became the new Table 1 and Table 3 and Table 4 became the new Table 2.

Round 2

Reviewer 1 Report

The Authors nicely and satisfactory responded to all my comments.

Reviewer 2 Report

Excellent paper. Accept in present form.

Reviewer 3 Report

The authors have improved the present manuscript considerably by incorporating suggestions and remarks by the reviewers. The figures and tables have been adequately revised in order to meet with the general concept of the study.